EMBO
Molecular Medicine

# Long-lived macrophage reprogramming drives spike protein-mediated inflammasome activation in COVID-19

Sebastian J Theobald[1,2,†], Alexander Simonis[1,2,†] [iD], Theodoros Georgomanolis[3] [iD], Christoph Kreer[4] [iD], Matthias Zehner[4], Hannah S Eisfeld[1,2], Marie-Christine Albert[5,6], Jason Chhen[1,2], Susanne Motameny[3] [iD], Florian Erger[3,7], Julia Fischer[1,2,8] [iD], Jakob J Malin[1,2] [iD], Jessica Gräb[1,2], Sandra Winter[1,2], Andromachi Pouikli[9], Friederike David[3] [iD], Boris Böll[1], Philipp Koehler[1,2,5], Kanika Vanshylla[4], Henning Gruell[4] [iD], Isabelle Suárez[1,8], Michael Hallek[1], Gerd Fätkenheuer[1,8], Norma Jung[1,8], Oliver A Cornely[1,2,5,8], Clara Lehmann[1,2,8], Peter Tessarz[5,9] [iD], Janine Altmüller[3], Peter Nürnberg[2,3], Hamid Kashkar[5,6], Florian Klein[4,8], Manuel Koch[10,11] & Jan Rybniker[1,2,8,*] [iD]

## Abstract

**Innate immunity triggers responsible for viral control or hyperinflammation in COVID-19 are largely unknown. Here we show that the SARS-CoV-2 spike protein (S-protein) primes inflammasome formation and release of mature interleukin-1β (IL-1β) in macrophages derived from COVID-19 patients but not in macrophages from healthy SARS-CoV-2 naïve individuals. Furthermore, longitudinal analyses reveal robust S-protein-driven inflammasome activation in macrophages isolated from convalescent COVID-19 patients, which correlates with distinct epigenetic and gene expression signatures suggesting innate immune memory after recovery from COVID-19. Importantly, we show that S-protein-driven IL-1β secretion from patient-derived macrophages requires non-specific monocyte pre-activation *in vivo* to trigger NLRP3-inflammasome signaling. Our findings reveal that SARS-CoV-2 infection causes profound and long-lived reprogramming of macrophages resulting in augmented immunogenicity of the SARS-CoV-2 S-protein, a major vaccine antigen and potent driver of adaptive and innate immune signaling.**

**Keywords** inflammasome; innate immunity; macrophage; NLRP3; SARS-CoV-2

**Subject Categories** Immunology; Microbiology, Virology & Host Pathogen Interaction

## Introduction

Since December 2019, coronavirus disease 2019 (COVID-19) has affected more than 140 million people globally (WHO, 2020). The disease is caused by infection with severe acute respiratory syndrome coronavirus 2 (SARS-CoV-2), a novel coronavirus. In COVID-19, little is known about protective or detrimental immune responses making rational therapeutic interventions difficult to assess. Subsets of patients fail to control viral replication some of which present with severe pneumonia, signs of hyperinflammation, and excessive release of cytokines in a second phase of the disease (Mangalmurti & Hunter, 2020; Mehta *et al*, 2020). This second phase immune response represents a putative target for host-directed therapeutic interventions and several approaches such as use of corticosteroids, blockade of the interleukin-6 receptor, inhibition of interleukin-1, or Janus kinases are currently being tested in clinical trials (Mehta *et al*, 2020). Despite a growing body of clinical data

1 Department I of Internal Medicine, Faculty of Medicine and University Hospital of Cologne, University of Cologne, Cologne, Germany
2 Faculty of Medicine and University Hospital of Cologne, Center for Molecular Medicine Cologne (CMMC), University of Cologne, Cologne, Germany
3 Faculty of Medicine and University Hospital of Cologne, Cologne Center for Genomics (CCG), University of Cologne, Cologne, Germany
4 Laboratory of Experimental Immunology, Institute of Virology, Faculty of Medicine and University Hospital of Cologne, University of Cologne, Cologne, Germany
5 Excellence Cluster on Cellular Stress Responses in Aging-Associated Diseases (CECAD), University of Cologne, Cologne, Germany
6 Institute for Medical Microbiology, Immunology and Hygiene (IMMIH), University of Cologne, Cologne, Germany
7 Faculty of Medicine, Institute of Human Genetics, University Hospital Cologne, Cologne, Germany
8 German Center for Infection Research (DZIF), Partner Site Bonn-Cologne, Cologne, Germany
9 Max Planck Research Group "Chromatin and Ageing", Max Planck Institute for Biology of Ageing, Cologne, Germany
10 Medical Faculty, Institute for Dental Research and Oral Musculoskeletal Biology, University of Cologne, Cologne, Germany
11 Medical Faculty, Center for Biochemistry, University of Cologne, Cologne, Germany
*Corresponding author. Tel: +49 221 478 89611; E-mail: jan.rybniker@uk-koeln.de
†These authors contributed equally to this work

supporting immunosuppressive treatment, knowledge on triggers of the SARS-CoV-2-specific inflammatory response and key cytokines that are involved is scarce (Vabret et al, 2020). Previous data could show that the major pro-inflammatory cytokine Interleukin-1-beta (IL-1β) is elevated in plasma from hospitalized COVID-19 patients and its associated signaling pathway seems to drive SARS-CoV-2 pathogenicity (Cavalli et al, 2020; Huet et al, 2020; Rodrigues et al, 2021).

IL-1β secretion is primarily initiated by inflammasomes that represent multiprotein signaling platforms responsible for the coordination of the early antimicrobial host defense (Broz & Dixit, 2016). Inflammasomes are assembled by pattern-recognition-receptors such as the NOD-, LRR-, and pyrin domain-containing protein 3 (NLRP3) following the detection of pathogenic microorganisms or danger signals in the cytosol of host cells. Upon activation, these receptors initiate the oligomerization of the adaptor protein ASC, serving as an activation platform for caspase-1. Active caspase-1 in turn cleaves pro-IL-1β yielding the mature active IL-1β, which can subsequently be secreted. NLRP3 inflammasome activation is a two-step process. In a priming step, cellular receptors recognize conserved pathogen-associated molecular patterns (PAMPs) leading to pro-IL-1β and pro-IL-18 expression. The activation step required for inflammasome assembly and secretion of mature IL-1β is triggered by a range of intrinsic or pathogen-derived stimuli such as (adenosine triphosphate) ATP, microbial toxins (e.g., nigericin), nucleic acids, or vaccine adjuvants (Mariathasan et al, 2006; Eisenbarth et al, 2008; Broz & Dixit, 2016; Swanson et al, 2019). For SARS-CoV-2, priming triggers are unknown. However, previous studies performed with other pathogenic viruses have identified viral envelope glycoproteins as potent inducers of innate immune activation and inflammatory cytokine production (Kurt-Jones et al, 2000; Boehme & Compton, 2004; Dosch et al, 2009; Escudero-Perez et al, 2014; Olejnik et al, 2018). Here, we show that the spike glycoprotein (S-protein), a major SARS-CoV-2 antigen and focus of therapeutic strategies and vaccine design, initiates inflammasome activation and IL-1β secretion selectively in pre-activated patient-derived human macrophages.

## Results

### The SARS-CoV-2 S-protein selectively triggers IL-1β secretion in patient-derived macrophages

The SARS-CoV-2 S-protein is a surface exposed and highly immunogenic viral component which induces a potent humoral immune response. In order to study the role of the SARS-CoV-2 S-protein in innate immune signaling, we first affinity-purified the protein using a HEK293 expression system (Fig 1A) (Wrapp et al, 2020). The S-protein specifically bound COVID-19 patient-derived IgG but not IgG from SARS-CoV-2 naïve controls confirming selective reactivity with patient-derived antibodies (Fig 1B). Next, we isolated peripheral blood mononuclear cells (PBMC) from hospitalized COVID-19 patients with moderate or severe disease and from SARS-CoV-2 naïve healthy controls followed by positive selection of CD14+ monocytes which were differentiated to macrophages by incubation with M-CSF (patient characteristics are provided in the Appendix Table S1 and Appendix Fig S1A). Overall monocyte counts and phenotypes were similar in both groups whereas plasma IL-1β levels were significantly higher in COVID-19 patients compared to healthy, SARS-CoV-2 naïve controls (Appendix Fig S1B–D). Isolated macrophages were subsequently exposed to 0.1 μg/ml SARS-CoV-2 S-protein followed by addition of nigericin or ATP as the inflammasome activating signal (Fig 1C). We show that the S-protein potently triggers secretion of IL-1β into the cell supernatants of patient-derived macrophages after sequential incubation with nigericin (Fig 1D). Intriguingly, cells from SARS-CoV-2 naïve healthy controls were non-reactive toward the S-protein (Fig 1D, Appendix Fig S1E). Even increasing the S-protein concentration by 100-fold to 10 μg/ml had little stimulatory effect on IL-1β secretion of macrophages derived from SARS-CoV-2 naïve donors (Fig EV1A). In contrast to the S-protein, lipopolysaccharide (LPS), a classical PAMP capable of priming inflammasome activation, induced IL-1β secretion in both groups when combined with nigericin, indicating functional inflammasome signaling pathways (Fig 1D). LPS and nigericin treatment led to higher IL-1β levels in supernatants of macrophages from COVID-19 patients. However, LPS/nigericin treatment of macrophages from naïve controls clearly led to secretion of significant amounts of IL-1β, which was not the case for S-protein stimulated naïve cells (Fig 1D). Macrophage treatment with LPS, nigericin, or S-protein alone had no effect on IL-1β secretion showing that the S-protein solely functions as an NLRP3-inflammasome priming signal requiring a second stimulus for IL-1β secretion (Figs 1D and EV1B). In patient-derived macrophages, IL-1β secretion was also induced following stimulation with the S-protein and the alternative signal 2 inducer ATP (Fig EV1C). Finally, we quantified tumor necrosis factor alpha (TNFα), a cytokine, which is secreted independently from the NLRP3 inflammasome. Macrophage stimulation with the S-protein led to secretion of TNFα in both patient-derived cells and SARS-CoV-2 naïve cells (Fig 1E).

**Figure 1. SARS-CoV-2 Spike protein induces IL-1β secretion in macrophages from COVID-19 patients.**

A SDS–PAGE of the recombinant SARS-CoV-2 spike protein (S-protein).

B SARS-CoV-2 spike binding assay of IgGs isolated from COVID-19 patients (n = 3; red circles) or SARS-CoV-2 naïve individuals (SC-naïve) (n = 2; blue circles).

C Experimental scheme: After PBMCs isolation, CD14+ cells were enriched by positive selection. Subsequently, 5 × 10^4 CD14+ cells/well were seeded and incubated in the presence of M-CSF for 5 days. Differentiated macrophages were stimulated with/without recombinant SARS-CoV-2 spike protein or lipopolysaccharide (LPS) for 4 h. To activate IL-1β secretion nigericin or ATP was added for 2 h. Finally, IL-1β secretion was quantified by ELISA.

D Quantification of IL-1β concentration (pg/ml) in the supernatant of primary macrophage cultures from COVID-19 patients (n = 44; red bars) or SC-naïve individuals (n = 24; blue bars) stimulated with LPS or S-protein (0.1 μg/ml). To activate IL-1β secretion, nigericin was added for 2 h.

E TNF-α (pg/ml) secreted from macrophages derived from SC-naïve and COVID-19 patients as described above. Cells were stimulated with LPS (SC-naïve/COVID-19 n = 4), LPS with nigericin (SC-naïve n = 4; COVID-19 n = 6), S-protein (SC-naïve n = 5; COVID-19 n = 4), S-protein with nigericin (SC-naïve n = 3; COVID-19 n = 4) or left unstimulated (SC-naïve n = 3; COVID-19 n = 4). Significances shown in the figure are always in comparison with the unstimulated control.

Data information: Graphs show mean ± SEM. **P < 0.01; ****P < 0.0001. For statistical analysis, two-way ANOVA with Tukey post hoc test was used.

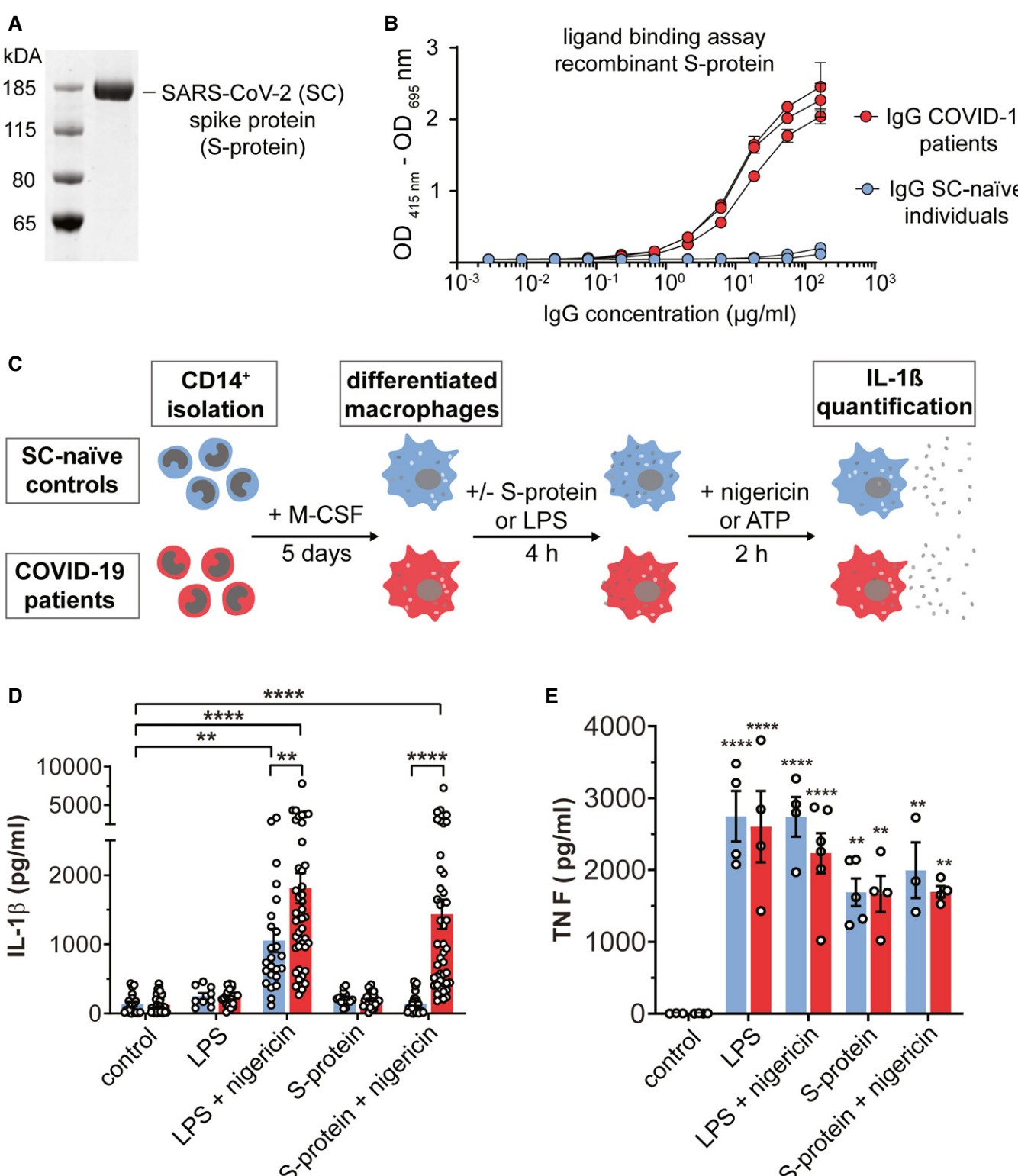

**Figure 1.**

**Full-length S-protein positively regulates NLRP3 and initiates secretion of mature IL-1β**

To confirm inflammasome activation in COVID-19 patient-derived macrophages upon S-protein priming, we first used a chemical genetics approach and pre-treated COVID-19 patient-derived macrophages with MCC950, a selective NLRP3 inhibitor (Coll *et al*, 2015), which efficiently blocked IL-1β secretion in S-protein/nigericin-stimulated patient cells (Fig 2A). Similar results were obtained when treating cells with VX765, a caspase-1 inhibitor (Fig 2A).

Pretreatment with these inhibitors blocked IL-1β in LPS stimulated patient cells as well (Fig EV1D).

We next quantified NLRP3 protein levels in macrophage cell lysates using immunoblot analysis. NLRP3 was upregulated in LPS exposed cells from both COVID-19 patients and naïve controls. In contrast to LPS, S-protein stimulation led to increased NLRP3 levels in patient cells only which is in line with our findings on IL-1β secretion (Figs 2B and EV1E). Using immunofluorescence microscopy of human macrophages, we also show ASC speck formation, a micrometer-sized structure formed by the adaptor protein ASC (apoptosis-associated speck-like protein containing a CARD) which is a hallmark of inflammasome activation. Intriguingly, ASC specks were identified at higher frequencies in S-protein/nigericin-treated macrophages from COVID-19 patients compared to treated cells from naïve individuals (Fig 2C and D).

Gene expression analyses by quantitative real-time PCR showed that the S-protein also significantly increased IL-1β mRNA levels in macrophages derived from COVID-19 patients, and, to a lesser extent, from SARS-CoV-2-naïve controls (Fig 2E). Notably, baseline IL-1β mRNA levels were higher in COVID-19 patient-derived cells indicating that differential regulation of IL-1β secretion in patients versus SARS-CoV-2 naïve controls occurs on the transcriptional level (Fig 2E). Similar findings were made for the cytosolic IL-1β precursor protein (pro-IL-1β) which was quantified by immunoblot (pro-IL-1β) and ELISA (total IL-1β) in S-protein treated cells (Figs 2F and EV1F and G).

We also quantified NEK7, a member of the family of mammalian NIMA-related kinases in macrophage lysates. NEK7 is an essential component of the NLRP3 inflammasome (He et al, 2016; Shi et al, 2016). We detected higher levels of NEK7 in macrophages from COVID-19 patients compared to levels found in macrophages from SARS-CoV-2 naïve individuals (Fig EV1H and I). LPS or S-protein stimulation for 4 h did not further increase NEK7 levels, which is in-line with observations made by others (He et al, 2016; Shi et al, 2016).

Proteolytic cleavage of inactive pro-IL-1β and subsequent secretion of active IL-1β requires inflammasome-dependent caspase-1 activity. Accordingly, we detected cleaved/mature IL-1β in supernatants of macrophage cultures using immunoblot analysis. LPS/nigericin treatment led to secretion of cleaved IL-1β from both patient-derived macrophages and macrophages from SARS-CoV-2 naïve individuals whereas S-protein/nigericin treatment selectively induced secretion of cleaved IL-1β in patient-derived cells only (Fig 2G). Another substrate of active caspase-1 is gasdermin D (GSDMD), a pore-forming protein, which induces necrotic cell death via pyroptosis. Cleaved GSDMD-N was detected in lysates of S-protein/nigericin-treated cells with a pattern comparable to cleaved IL-1β detected in macrophage supernatants (Fig 2H). To confirm S-protein/nigericin-induced cell death, we also quantified dead cells in S-protein/nigericin-treated and non-treated cells using 7-AAD fluorescent staining which showed a significantly higher proportion of dead cells (7-AAD$^+$) in S-protein-treated cells (Fig EV1J).

To better define structural requirements for S-protein primed IL-1β secretion, we tested the affinity-purified S2 membrane-fusion domain in our ex vivo assay as signal-1 molecule. This subunit combined with nigericin failed to induce IL-1β secretion indicating that the full S-protein trimer is required for immune signaling (Fig EV1K). As further confirmation for an S-protein specific effect, we affinity-purified a SARS-CoV-2 accessory protein of unknown function (ORF8) in the same expression system and tested this control protein in our assay. As expected, ORF8 failed to potentiate IL-1β secretion, as signal 1 protein, in patient-derived macrophages (Fig EV1L).

We also assessed whether post-transcriptional modification of NLRP3 derived from patient or SARS-CoV-2 naïve macrophages may explain differential inflammasome regulation. Higher levels of NLRP3 ubiquitination have been shown to abrogate inflammasome formation (Py et al, 2013). However, we were not able to detect different NLRP3 ubiquitination pattern in macrophages from diseased

---

**Figure 2. SARS-CoV-2 spike protein primes NLRP3 inflammasome and induces pyroptosis.**

A  Chemical inhibition of IL-1β (pg/ml) secretion after pre-incubation of macrophages from COVID-19 patients (red) for 2 h with DMSO (n = 21), MMC950 (n = 21), or VX-765 (n = 8), followed by 4 h incubation with S-protein and additional 2 h with nigericin. For statistical analysis, one-way ANOVA with Tukey post hoc test was used.

B  Western blot analysis of total cell lysates of macrophages from a SARS-CoV-2 naïve individual (SC-naïve) (top) and COVID-19-infected individual (bottom). Macrophages were stimulated with LPS or S-protein (4 h) followed by 2 h of nigericin treatment. Control cells were left unstimulated. Antibodies against NLRP3 (110 kDa) and β-actin (loading control) were used for detection. Representative example of three individual experiments.

C  Representative confocal microscope pictures of macrophages from SC-naïve (top) and COVID-19 patients (bottom). Nuclei are stained with DAPI (blue) and ASC speck are labeled by immunofluorescence staining (green). Macrophages were stimulated with LPS (left) and S-Protein (right) for 4 h followed by stimulation with nigericin in both cases. Pictures were taken with a 60× objective using the same microscope settings.

D  Quantification of ASC speck from SC-naïve (n = 3, blue) and COVID-19 patients (n = 8, red) derived macrophages. Percentage of ASC specks was calculated per DAPI stained nuclei. For statistical analysis, two-way ANOVA with Tukey post hoc test was used.

E  IL-1β gene expression of macrophages from COVID-19 patients (n = 5; red bars) or SC-naïve individuals (n = 3 (S-protein) n = 4 (control); blue bars) stimulated with/without S-Protein were determined by qRT–PCR. Data are normalized to β-actin. Statistical significance was analyzed using the Kolmogorov–Smirnov test.

F  Western blot analysis of total cell lysates of macrophages from SARS-CoV-2 naïve individuals (SC-naïve) (top) and COVID-19-infected individuals (bottom). Macrophages were stimulated with LPS or S-protein (1 h). Control cells were left unstimulated. Antibodies against pro-IL-1β (31 kDa) and β-actin (loading control) were used for detection. Representative example of three experiments performed with individual patients and SARS-CoV-2 naïve controls.

G  Western blot analysis comparing macrophage precipitated supernatants from a SC-naïve (top) and a COVID-19-positive patient (bottom). Macrophages were treated with DMSO or MCC950 (2 h) and further stimulated with LPS or S-protein (4 h) followed by 2 h of nigericin treatment. Antibodies binding cleaved IL-1β (17 kDa) were used for detection. Representative example of three individual experiments.

H  Western blot analysis of total cell lysates of macrophages from a SARS-CoV-2 naïve individuals (SC-naïve) (top) and COVID-19-infected individual (bottom). Macrophages were stimulated with LPS or S-protein (4 h) followed by 2 h of nigericin treatment. Control cells were left unstimulated. Antibodies targeting cleaved GSDMD (110 kDa) and β-actin (loading control) were used for detection. Representative example of three individual experiments.

Data information: Graphs show mean ± SEM. *P < 0.05; **P < 0.01; ****P < 0.0001.
Source data are available online for this figure.

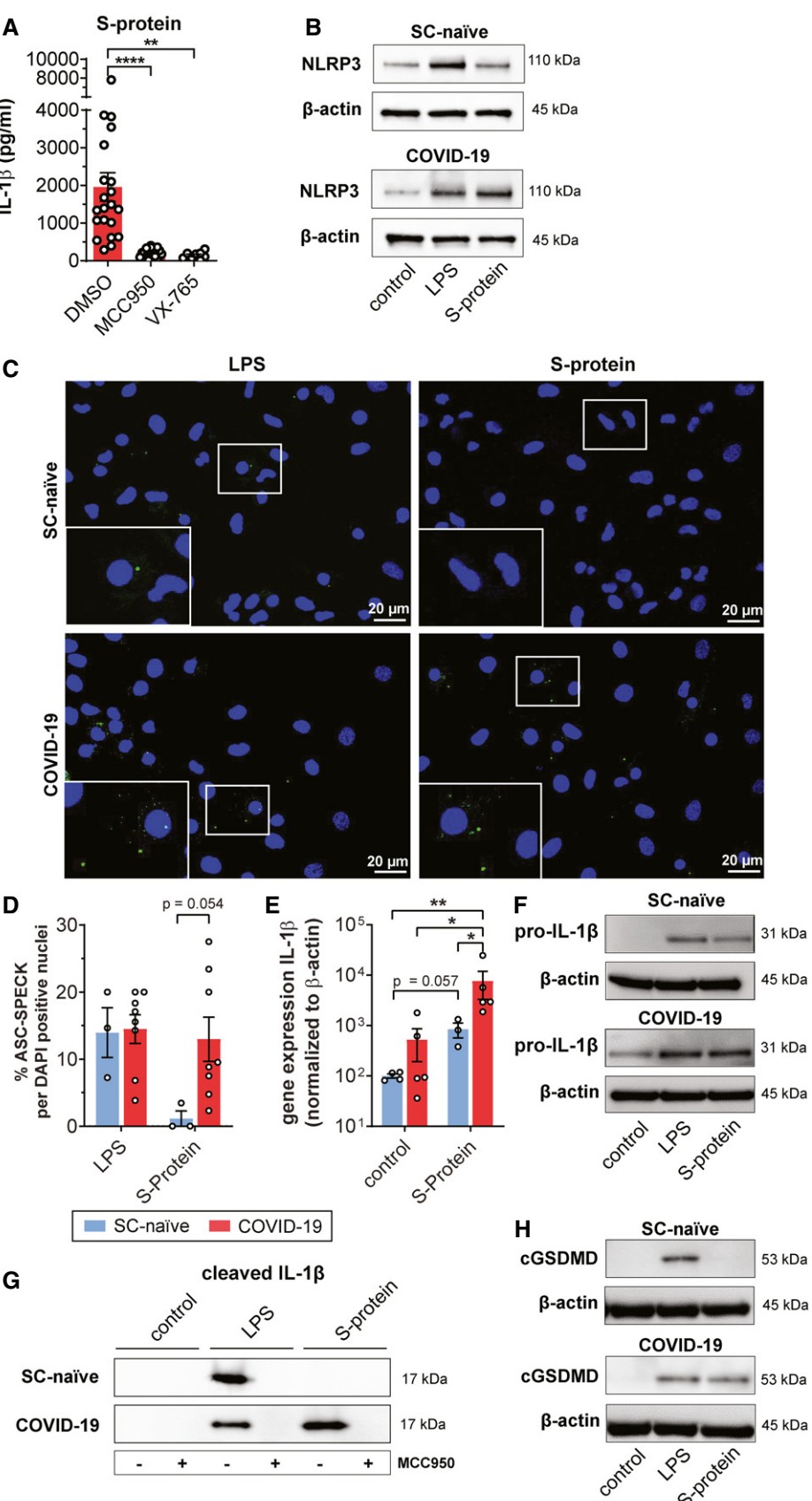

**Figure 2.**

and SARS-CoV-2 naïve controls (Appendix Fig S2A and B). Ubiquitinase isopeptidase inhibitor-treated cells were used as controls (Py et al, 2013). We also determined the phosphorylation status of NLRP3 since phosphorylation of this protein has been linked to its ubiquitination. Using a phospho-NLRP3 antibody, we did not detect differential phosphorylation of NLRP3 derived from patient or naïve macrophages (Appendix Fig S2C) Together, these data suggest that in COVID-19, differential regulation of the macrophage NLRP3 inflammasome rather occurs on the transcriptional level.

## S-protein-mediated IL-1β secretion in mildly diseased and convalescent COVID-19 patients

We revisited our finding of differential macrophage reactivity in SARS-CoV-2 patients compared to those from SARS-CoV-2 naïve individuals and performed a sub-analysis of COVID-19 patients with mild disease symptoms and convalescent individuals (patient characteristics can be found in Appendix Table S1). Interestingly, macrophages from outpatients with mild disease or asymptomatic infection showed elevated IL-1β secretion upon stimulation similar to hospitalized patients with severe disease indicating that macrophage reactivity toward the S-protein does not correlate with disease severity (Fig 3A). While some individuals of our outpatient cohort were SARS-CoV-2 PCR positive, plasma IgG negative and presented with mild symptoms upon sampling for our study, others were already PCR negative, seroconverted, and without any symptoms (convalescent patients) when analyzed. Intriguingly, macrophages from these convalescent patients strongly secreted IL-1β upon S-protein stimulation. Two convalescent individuals were tested sequentially, 18, 26, and 56 days after diagnosis. IL-1β levels upon S-protein stimulation gradually declined over time while IL-1β levels of LPS-treated cells remained stable indicating SARS-CoV-2-specific innate immune memory after recovery, which wanes over time (Fig 3B).

## Distinct gene expression signatures in S-protein stimulated macrophages

Intrigued by our observation of S-protein priming of the inflammasome selectively in patient-derived macrophages, we performed RNA-Seq-based gene expression analyses in LPS or S-protein stimulated and unstimulated macrophages from SARS-CoV-2 naïve (SC-naïve) and convalescent (SC-conv) individuals (Dataset EV1). All transcriptomic data were generated in cells that had not been exposed to nigericin. Macrophage RNA-Seq data of four additional, fully recovered convalescent COVID-19 patients with normalized C-reactive protein and detectable SARS-CoV-2-specific plasma-IgG were matched with data from SARS-CoV-2 naïve controls (Appendix Table S1). Of note, despite being isolated from fully recovered patients, macrophages were still responsive to S-protein priming leading to strong IL-1β secretion (Fig 3C). Mean time span between COVID-19 diagnosis and blood sampling for our analysis was 36 days with a maximum of 49 days (7 weeks) in one patient (Appendix Table S1).

On a global scale, both S-protein and LPS led to considerable shifts in gene expression of macrophages from SC-naïve and SC-conv individuals. While LPS stimulation led to similar numbers of differentially expressed genes (DEGs) in both groups (SC-naïve:

7,415; SC-conv: 8,306 (Padj < 0.05)), there was a marked difference of 2,445 genes (50.5%) which were up- or down-regulated upon S-protein stimulation in convalescent versus naïve individuals, respectively (SC-naïve: 2,497, SC-conv: 4,942) (Fig 3D). This clearly indicates that, in recovered COVID-19 patients, S-protein-specific differential transcriptional regulation occurs on a much higher level compared to LPS-driven differential expression (Dataset EV1A–D, Fig 3D).

Principal component analysis (PCA) confirmed distinct clustering of all three experimental groups (S-Protein, LPS, and unstimulated) (Fig EV2A). Further, SC-naïve and SC-conv clustered differentially when stimulated with S-protein or LPS (Fig EV2A).

We then performed KEGG pathway enrichment analyses of DEGs primarily focusing on SC-conv individuals after different modes of stimulation. The datasets largely confirm that the S-protein is a potent inducer of innate immunity-associated gene expression and pro-inflammatory pathways (Fig 3E, Table 1, Fig EV2B and C). Key components of NOD-like receptor and inflammasome signaling were highly upregulated in S-protein stimulated cells compared to non-treated cells (NLRP3 (6.8 fold), caspase-1 (3.4 fold) as well as IL-1β (341.7 fold), IL-18 (4.5 fold) and their respective receptors) (Fig 3F, Table 1, Dataset EV1D). In addition, NRLP3-inflammasome independent cytokine mRNAs (e.g., IL-6, IL-12 IL-15, IL-16, IL-36, TNF) and several chemokine receptors were positively regulated in response to the S-protein (Dataset EV1D).

While there was considerable overlap in genes regulated by both LPS and the S-protein, a total of 3,994 genes (Padj < 0.05) were differentially expressed by either LPS or the S-protein clearly showing a distinct signature for these two antigens (Fig 3G, Appendix Fig S3A, Dataset EV1E).

Both PAMPs strongly activated genes associated with nuclear factor kappa-light-chain-enhancer of activated B cells (NF-κB) signaling (Dataset EV1B and D). NF-κB is a master regulator coordinating transcription of inflammation-associated genes upon activation of toll-like receptor (TLR) signaling. LPS primarily signals via TLR4 and we found that the associated gene is upregulated in LPS treated but not in S-protein-treated macrophages (Dataset EV1E). The S-protein led to pronounced upregulation of TLR2 primarily in macrophages derived from convalescent individuals (Fig 3F). We confirm relevance of this finding by quantifying TLR2 on the cell surface of CD14[+] macrophages using flow cytometry analysis (FACS). Patient-derived cells showed significantly higher TLR2 levels compared to SARS-CoV-2-naïve controls (Fig 4A). In contrast, quantification of the cell surface receptor ACE2 by FACS analysis revealed no difference in patient-derived macrophages compared to macrophages isolated from naïve individuals (Fig EV3A). Treatment with MMG-11, a selective small molecule antagonist of TLR2, abrogated IL-1β secretion in S-protein-stimulated macrophages (Fig 4B) but failed to block LPS induced IL-1β secretion (Fig 4C). Zymosan, a well-described TLR2 agonist and priming molecule of the NLRP3 inflammasome, was used as a control. IL-1β secretion of zymosan-stimulated macrophages could be blocked by MMG-11 treatment (Fig 4D). Of note, macrophages isolated from naïve individuals secreted IL-1β upon priming with zymosan which stands in contrast to S-protein treated cells (Fig 4D). KINK-1, a selective chemical inhibitor of NF-κB, was used as a control substance which blocked IL-1β secretion in all conditions tested (Fig 4B–D). We also performed TLR2 blocking experiments using neutralizing monoclonal

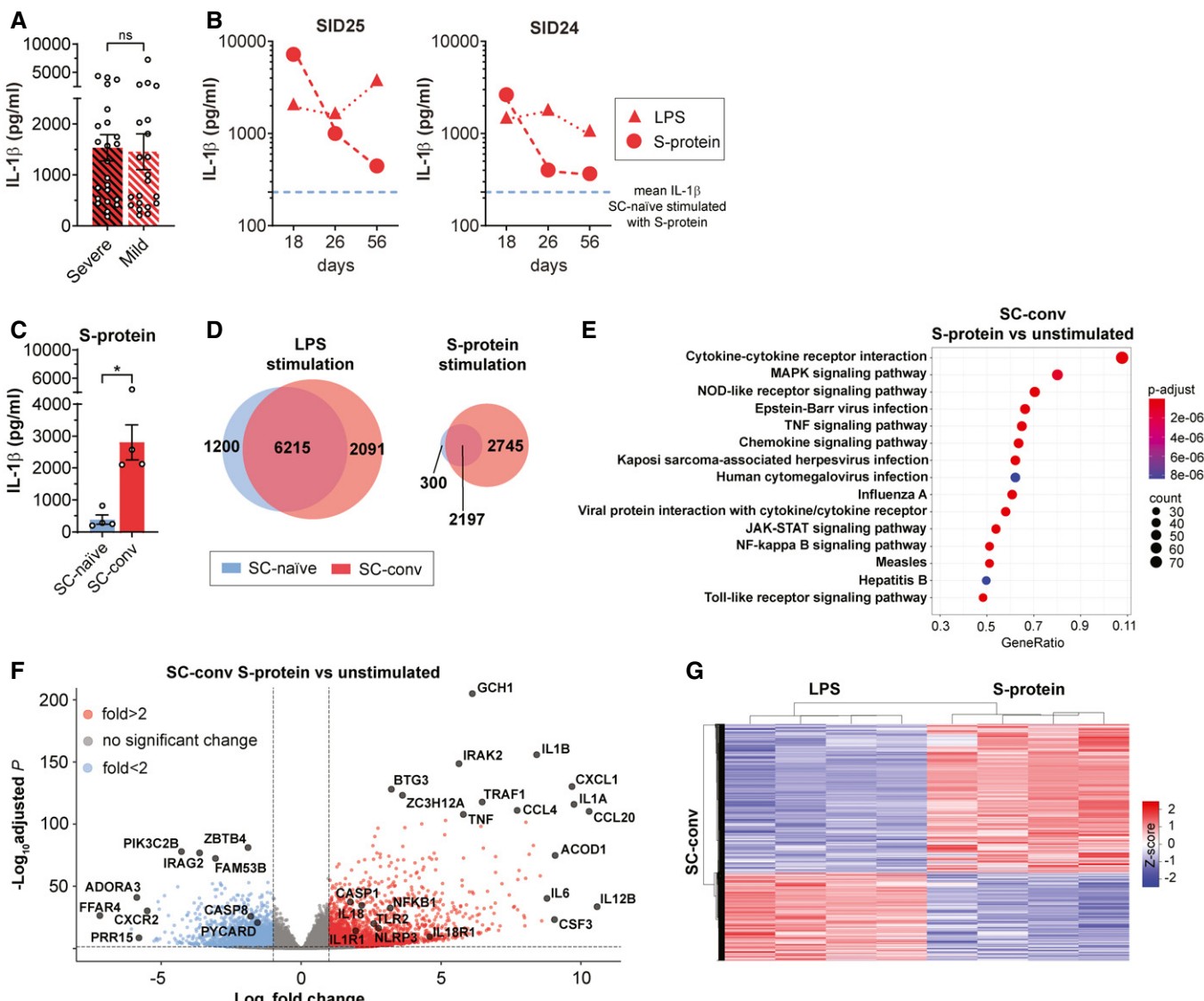

**Figure 3. S-protein stimulated macrophages from convalescent individuals secrete IL-1β and show distinct gene expression signatures.**

A   Subgroup analysis of IL-1β secretion of macrophages after stimulation with S-protein and nigericin from COVID-19 patients with severe (red/black; *n* = 23) or mild (white/red; *n* = 21) disease. Student's *t*-test with Welsh corrections was used to calculate statistical differences. Graph shows mean ± SEM.

B   CD14[+] cells of two convalescent individuals were isolated sequentially at d18, d26, and d56 after confirmation of SARS CoV2 infection by PCR. Following M-CSF differentiation, macrophages were stimulated with LPS (red triangle) or S-Protein (red circles) and IL-1β concentration (pg/ml) was quantified upon incubation with nigericin. Blue dashed line indicates mean IL-1β level of SC-naïve individuals stimulated with S-protein and nigericin.

C   Quantification of IL-1β concentration (pg/ml) in the supernatant of primary macrophage cultures from convalescent COVID-19 patients (SC-conv) (*n* = 4; red bars) or SARS-CoV-2 naïve individuals (SC-naïve) (*n* = 4; blue bars) stimulated with S-protein (1 μg/ml) and nigericin. Students *t*-test with Welsh corrections was used to calculate statistical differences as indicated. Graph shows mean ± SEM. *P < 0.05.

D   Venn diagram showing number of differentially expressed genes (DEGs) in healthy/naïve (SC-naïve; blue) and convalescent COVID-19 patient-derived macrophages (SC-conv; red) upon stimulation with LPS (left) or S-Protein (right). Circle sizes of the Venn diagram correspond to the number of genes.

E   KEGG-gene enrichment analysis, based on RNA-Seq data, showing activated signaling pathways in S-protein stimulated macrophages compared to unstimulated in convalescent COVID-19 patient-derived macrophages (SC-conv). *P*adj values are indicated (color code), and size of the circles represents number of DEGs (count). *P*-values are calculated using Wald test and *P*-adjusted values using the FDR/Benjamini–Hochberg approach.

F   Volcano plot showing DEGs in S-protein stimulated macrophages from convalescent COVID-19 patients (SC-conv) compared to unstimulated cells. Negative log₁₀ adjusted *P*-values are plotted against the log₂-fold change. Select genes are labeled. *P*-values are calculated using Wald test and *P*-adjusted values using the FDR/Benjamini–Hochberg approach.

G   Heat map indicating DEG patterns comparing LPS or S-protein stimulation of SC-conv macrophages. *Z*-score is indicated in a color score.

Data information: RNA-Seq analyses included data with *P*adj < 0.05 and log₂-fold change of ≤−1 and ≥1.

Table 1. Summary of selected macrophage RNA-Seq and miRNA data generated in this study. Key pathways and key genes differentially regulated in different groups and conditions. Full datasets have been uploaded as supplementary tables.

| Group | Condition | DEGs | Key pathways | Key genes | miRNA | Linked figures and datasets |
|---|---|---|---|---|---|---|
| SC-conv | SP versus unstim | 2445 | NOD-like signaling; inflammatory responses | *IL-1β*; *NLRP3*; *CASP1* | ND | Fig 3E and F, Dataset EV1D |
| SC-conv | LPS versus SP | 891 | Interferon signaling; Response to LPS | *MAP3K14*; *IFI27/35*; *IRF4* | ND | Fig 3G, Dataset EV1E, Appendix Figs S3A and B |
| SC-conv versus SC-naïve | SP | 1802 | cytokine signaling; inflammatory responses | *NLRP3*; *CASP1* | ND | Fig 5A and B, Dataset EV1F |
| SC-conv versus SC-naïve | LPS | 1157 | No inflammatory pathway differentially regulated | / | ND | Dataset EV1G |
| SC-conv versus SC-naïve | unstim. | 1312 | metal ion homeostasis; cellular oxidation-reduction process; NF-κB signaling | *MT1A*; *S100A8* | mir-155; −221; −222 | Fig 5C and D, Fig 6A, Dataset EV1H and I |

DEGs, differentiated expressed genes; LPS, lipopolysaccharide; SC-conv, COVID-19 convalescent; SC-naïve, SARS-CoV-2 naïve; SP, S-protein; Unstim, unstimulated; ND, not determined.

antibodies directed against this receptor. Anti-TLR2 antibody treatment of macrophages reduced S-protein/nigericin-dependent IL-1β secretion in a dose-dependent manner giving further evidence for a role of TLR2 in S-protein signaling, whereas LPS-dependent IL-1β secretion was not affected by the blocking antibody (Fig 4E).

Pathway analyses of gene networks activated exclusively by LPS were mainly associated with interferon (IFN) signaling (Appendix Fig S3B, Table 1). LPS exposure is known to positively regulate both type I interferons and IFN-γ-associated genes in macrophages as also detected in our analysis (Appendix Fig S3A and B, Dataset EV1A,B) (Fultz *et al*, 1993; Sheikh *et al*, 2014). Interestingly, despite being strongly linked with the antiviral host response, transcripts of these important signaling molecules were not activated by the S-protein (Dataset EV1D).

Together these data show that the S-protein potently induces NOD- and TLR-associated inflammatory pathways in convalescent COVID-19 patient-derived macrophages with a transcriptomic signature, which is distinct from the signature found in LPS-treated cells.

### Transcriptomic and epigenetic changes in macrophages from convalescent individuals associated with innate immune memory

Next, we compared DEGs in S-protein stimulated SC-conv macrophages to those of S-protein stimulated SC-naïve macrophages. Upon priming with the S-protein, a total of 1,802 genes were significantly up- or down-regulated in convalescent COVID-19 patients only (Fig 5A, Table 1, Dataset EV1F), strongly indicating a differential response in the two groups upon stimulation with the same antigen. Among the inflammatory genes upregulated in S-protein stimulated SC-conv macrophages but not in S-protein stimulated SC-naïve macrophages were NLRP3 and caspase-1 (Dataset EV1F). Of note, these differences were not observed in LPS-treated cells where the two inflammasome-associated genes were equally upregulated in macrophages from both groups, which is fully in line with our observation of LPS-driven inflammasome activation independent from previous SARS-CoV-2 infection (Table 1, Fig 2B–H, Dataset EV1G). A gene ontology analyses (GO) of biological processes (BP) highlighted mainly inflammation and cytokine signaling-associated pathways significantly upregulated in S-protein treated SC-conv cells compared to SC-naïve cells (Fig 5B, Table 1, Fig EV3B and C).

Thus, our transcriptomics analysis confirms that SARS-CoV-2 naïve individuals fail to upregulate key inflammasome-associated genes upon S-protein exposure despite the fact that the transcriptional machinery is functional in these cells when switching to another PAMP (LPS/zymosan).

To determine which pathway contributes to the effect of selective S-protein induced IL-1β secretion, we focused on transcriptomic differences in SC-conv and SC-naïve macrophages, which had not been stimulated with LPS or the S-protein (Dataset EV1H). The lifespan of circulating classical human monocytes is < 24 h (Patel *et al*, 2017) and several cycles of monocyte renewal should have been occurred in the monocyte pool of our convalescent patients since recovery from COVID-19. Despite this fact, we found a surprisingly high number of 1,312 DEGs in the two groups (*P*adj < 0.05) (Dataset EV1H). There was a clear and highly significant pattern that allowed for separation of the SC-conv and SC-naïve group (Fig 5C).

When dissecting the corresponding networks, we identified significant changes primarily in genes associated with the cellular stress response. This includes metal ion homeostasis, the cellular oxidation–reduction process, and NF-κB signaling-associated genes (Fig 5D, Table 1). These pathways are strongly associated with the metabolic and transcriptional regulation of innate immunity and immune memory. In SC-conv macrophages, a large set of oxidoreductases were differentially expressed. Genes coding for metallothioneins (MT1A, MT1X, MT1M), metal-ion binding proteins which potently control oxidative stress by capturing superoxide and hydroxyl radicals, were the most down-regulated genes in macrophages isolated from convalescent COVID-19 patients (Subramanian Vignesh & Deepe, 2017) (Fig 5D). We speculated that chemically induced oxidative stress may render macrophages from COVID-19-naïve individuals responsive toward S-protein priming. To confirm this, we pre-treated macrophages with low amounts of $H_2O_2$ or FCCP to induce a shift in the intracellular redox balance. Interestingly, treatment with these substances and subsequent exposure to S-protein/nigericin led to IL-1β secretion to levels found in S-protein/nigericin-treated macrophages isolated from COVID-19 patients indicating that redox signaling contributes to selective priming of macrophages. $H_2O_2$ or FCCP treatment alone had no effect on IL-1β secretion (Fig 5E).

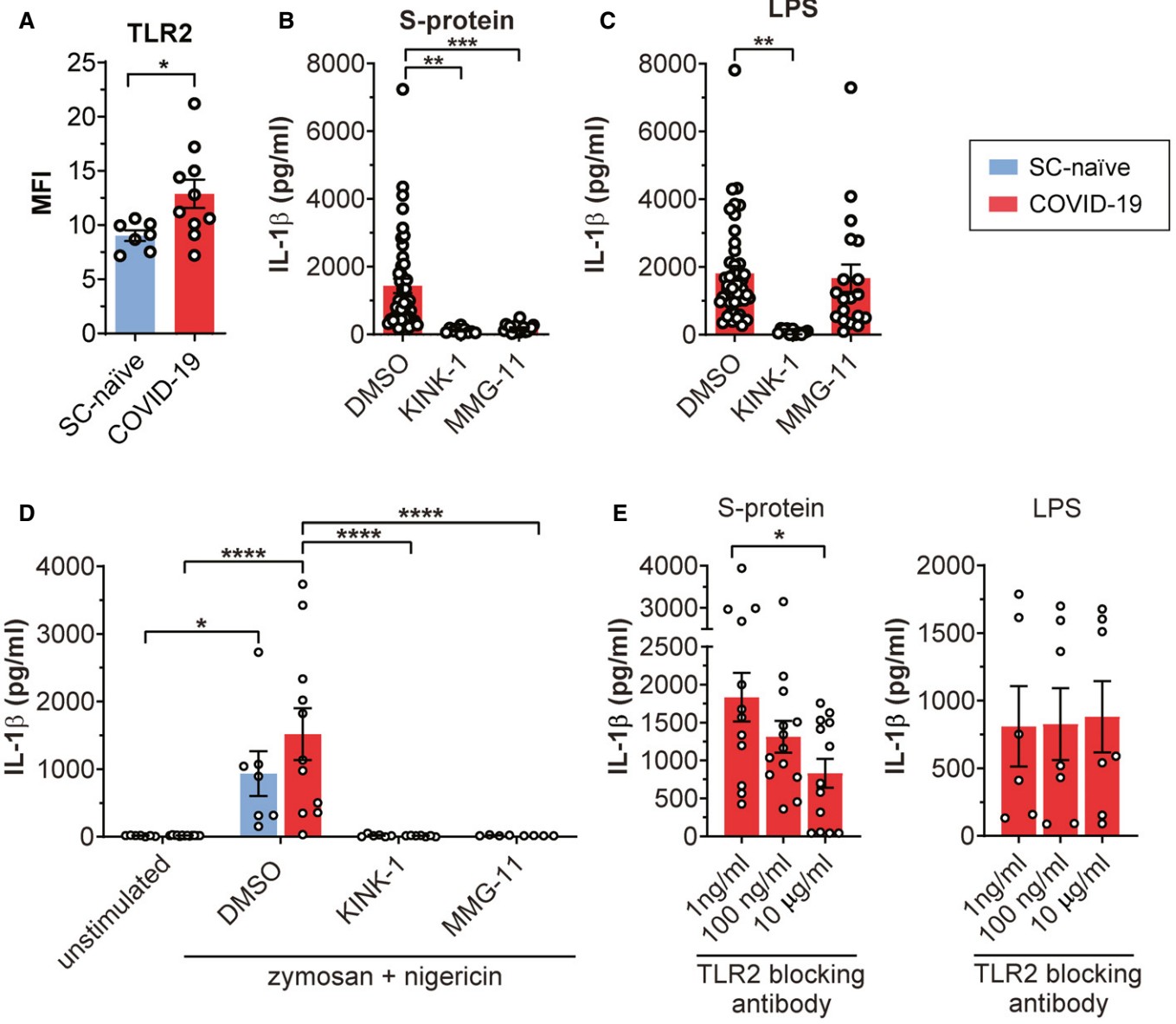

**Figure 4. TLR-2 is involved in IL-1β secretion of S-protein stimulated macrophages.**

A   Mean fluorescence intensity (MFI) comparing surface TLR2 expression on healthy/naïve (SC-naïve; blue; $n = 7$) and COVID-19 patient (SC-conv; red; $n = 10$) derived macrophages. Students $t$-test with Welsh corrections was used to calculate statistical differences as indicated.

B   IL-1β concentration (pg/ml) in supernatants of COVID-19 patient-derived macrophages pre-treated with DMSO ($n = 44$), KINK-1 ($n = 11$), and MMG-11 ($n = 18$) and subsequent stimulation with S-protein and nigericin. For statistical analysis, one-way ANOVA was used.

C   IL-1β concentration (pg/ml) in supernatants of COVID-19 patient-derived macrophages pre-treated with DMSO ($n = 44$), KINK-1 ($n = 11$), and MMG-11 ($n = 19$) and subsequent stimulation with LPS and nigericin. For statistical analysis, one-way ANOVA was used.

D   IL-1β concentration (pg/ml) in supernatants from macrophages of SC naïve (blue; $n = 7$) and COVID-19 patients (red; $n = 11$). Macrophages were pre-treated with DMSO (SC-naïve $n = 7$; COVID-19 $n = 11$), KINK-1 (SC-naïve $n = 6$; COVID-19 $n = 7$), and MMG-11 (SC-naïve $n = 4$; COVID-19 $n = 4$) for 2 h and then stimulated with zymosan (4 h) and nigericin (2 h) as indicated. For statistical analysis, two-way ANOVA with Tukey post hoc test was used.

E   IL-1β concentration (pg/ml) in supernatants from COVID-19 patients which were stimulated for 2 h with a blocking anti-TLR2 monoclonal antibody in different concentrations as indicated. Subsequently, macrophages were stimulated with LPS (1 ng/ml: $n = 6$; 10 and 100 ng/ml: $n = 7$) and S-Protein (1 ng/ml: $n = 12$; 10 and 100 ng/ml: $n = 13$) for 4 h and both groups for 2 h with nigericin. For statistical analysis, one-way ANOVA was used.

Data information: Graphs show mean ± SEM. *$P < 0.05$; **$P < 0.01$; ***$P < 0.001$; ****$P < 0.0001$.

Finally, several damage-associated molecular patterns (DAMP) of the S100-familiy (e.g., S100A8 and S100A12) were differentially expressed in SC-conv macrophages compared to SC-naïve macrophages. These low molecular weight calcium- and zinc-binding proteins are endogenous activators of the NRLP3 inflammasome and other pro-inflammatory pathways (Wang *et al*, 2018)

(Fig 5D). Thus, our transcriptomic study on macrophages derived from COVID-19 convalescent individuals shows a gene expression pattern that seems to allow for rapid and selective activation of the inflammasome and IL-1 signaling upon S-protein exposure.

Long-lived modifications of transcriptomic signatures and innate immune responses in short-lived circulating cells as observed in our study require functional cell reprogramming. This can be achieved by inducible regulatory molecules, such as microRNAs (miRNA), which shape different transcriptional programs and outcomes that characterize immune memory (Curtale et al, 2019). We quantified and compared miRNA levels in non-stimulated macrophages from SC-conv and SC-naïve individuals and identified a total of 30 miRNAs which were significantly differentially expressed in SC-conv cells ($P < 0.05$) (Dataset EV1I). PCA-clustering analysis

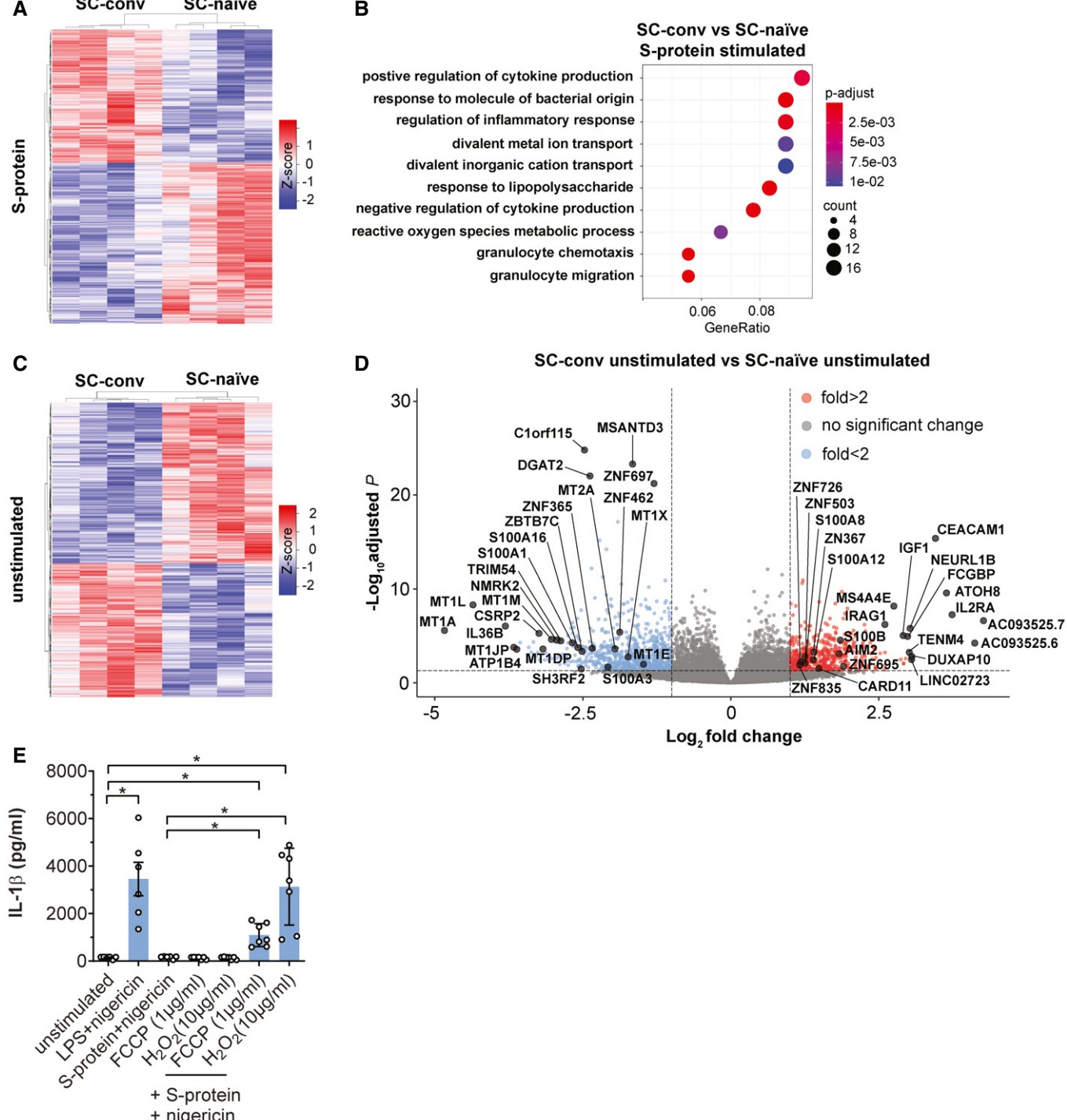

Figure 5.

**Figure 5. Transcriptomic changes are associate with innate immune responses and redox-status of stimulated macrophages.**

A   Heat map indicating DEG patterns comparing SC-conv and SC-naïve macrophages stimulated with the S-protein. Z-score is indicated in a color score.
B   BP enrichment analysis, based on RNA-Seq data, showing activated signaling pathways upon S-protein stimulation in SC-conv macrophages compared to SC-naïve macrophages. Padj values are indicated (color code) and circle size corresponds to the number of DEGs contributing to the pathways (count).
C   Heat map indicating DEG patterns comparing SC-conv and SC-naïve macrophages without any stimulation. Z-score is indicated in a color score.
D   Volcano plot showing DEGs in unstimulated COVID-19 patient-derived macrophages (SC-conv) compared to macrophages from SARS-CoV-2 naïve individuals (SC-naïve). Negative $\log_{10}$ Padj values are plotted against the $\log_2$-fold change. Selected genes of interest are labeled.
E   IL-1β concentration (pg/ml) in supernatants of SC-naïve individuals ($n = 7$) which were pre-treated for 24 h with FCCP (1 µg/ml) and $H_2O_2$ (10 µg/ml). Macrophages were then incubated with LPS or S-Protein (4 h) and nigericin (2 h) as indicated in the figure. Control cells were left unstimulated or stimulated with LPS or S-protein without pre-treatment with FCSP or $H_2O_2$ ($n = 6$). For statistical analysis, Brown–Forsythe and Welsh ANOVA test was used.

Data information: RNA-Seq analyses included data with Padj < 0.05 and $\log_2$-fold change of $\leq -1$ and $\geq 1$. P-values for the DEGs were calculated using Wald test and P-adjusted values using the FDR/Benjamini–Hochberg approach. For the pathways and GO analysis, P-values were calculated by hypergeometric distribution and the adjusted P-value was calculated with the Benjamini–Hochberg method. Graphs show mean ± SEM. *P < 0.05.

confirms distinct expression patterns of miRNAs in both experimental groups (Fig EV3D). Among those, we identified several miRNAs which are known targets of immune regulatory genes (Fig 6A). This includes miRNA-155 (mir-155) which is known to strongly control the inflammatory macrophage signature. Similarly, expression of mir-221 and mir-222, as observed in our study, has been associated with the host response to pathogens and innate immune memory (Furci *et al*, 2013; Seeley *et al*, 2018).

Finally, we confirmed epigenetic alterations of monocytes from convalescent individuals by mapping trimethylated H3K4 (H3K4me3) and acetylated H3K27 (H3K27ac) genome-wide using CUT&RUN (Skene & Henikoff, 2017). Epigenetic reprogramming of myeloid cells represents a hallmark of trained innate immunity (Netea *et al*, 2016). In general, active enhancers are marked by H3K27ac and transcription start sites are marked by H3K4me3 and H3K27ac (Kimura, 2013).

A global analysis of histone modifications associated with transcription start sites (TSS) revealed a difference in the overall coverage of specific loci in SC-conv versus SC-naïve cells with SC-conv samples showing lower occupancy overall (Fig 6B).

After peak-calling and identification of differentially enriched loci in the genomes of monocytes from SARS-CoV-2 naïve and convalescent COVID-19 patients, we performed functional enrichment analyses of nearby genes. We first focused on histone marks associated with potential enhancers as defined by the presence of H3K27ac and absence of H3K4me3. A gene ontology analysis of enriched sites in SC-conv monocytes revealed pathways primarily involved in inflammation-associated activation of myeloid cells and positive regulation of the immune response (Fig 6C). These pro-inflammatory pathways were not detected in samples derived from SC-naïve cells. The subsequently performed KEGG analysis of the enriched sites detected in SC-conv cells identified infectious disease-associated pathways (e.g., tuberculosis, *Escherichia coli* infection) and pathways associated with the antimicrobial response of macrophages (phagosome, lysosome) (Fig 6D). Intriguingly, many of the genes detected in this genome-wide histone modification analysis were also identified as upregulated in the RNA-seq experiments performed on S-protein stimulated macrophages or non-stimulated macrophages from convalescent individuals (e.g., IL-1β, IL-1α, MYD88, JAK1, CD14, TLR2, S100A8/9/12, NLRP3, Caspase 1) (Figs 6E and F, and EV4). Further, we identified enriched active promoters (H3K4me3- and H3K27ac-enriched peaks) for each condition. Here, the gene ontology analysis of SC-conv samples also revealed pathways primarily involved in

**Figure 6. Epigenetic control of immune memory in macrophages.**

A   Volcano plot showing differentially expressed miRNAs from unstimulated COVID-19 patient-derived macrophages (SC-conv) compared to those from naïve individuals (SC-naïve). Negative $\log_{10}$ adjusted P-values are plotted against the $\log_2$-fold change. Genes of interest are indicated. For miRNA analyses, data with a Padj value of smaller 0.05 and $\log_2$-fold change of $\leq -0.85$ (blue) and $\geq 0.85$ (red) were included.
B   Average profile plot showing distribution of histone modifications identified in CUT&RUN experiments displayed as normalized read counts per million around known transcription start site (TSS) of known genes. Color code for H3K4me3 (SARS-CoV-2 naïve/green; COVID-19/red) and for H3K27ac (SARS-CoV-2 naïve/yellow; COVID-19/purple).
C   GO biological process analysis performed on differentially enriched loci of H3K27ac peaks (enhancer) showing associated pathways from SC-conv monocytes compared to SC-naïve monocytes.
D   KEGG enrichment analysis performed on differentially enriched loci of H3K27ac peaks (enhancer) showing involved pathways from SC-conv monocytes compared to SC-naïve monocytes. Adjusted P-values are indicated (color code) and size of the circles corresponds to the number of detected genes associated with each pathway (count).
E   CNET plot (derived from analysis shown in D) illustrating genes involved in tuberculosis and phagosome KEGG pathways.
F   Representative gene loci (TLR2 (top) and S100A7A/8/9/12 (bottom) H3K27ac peaks) comparing SC-naïve (blue) and SC-conv (red) monocytes. Red peaks reach higher values than blue peaks (representative examples of four samples for each condition).
G   Quantification of IL-1β concentrations (pg/ml) in the supernatant of primary macrophage cultures from patients with active/untreated tuberculosis ($n = 6$; red/black bars) or healthy/SARS-CoV-2 naïve individuals ($n = 3$; blue bars) or the same patients after 6 months of anti-tuberculous treatment; $n = 6$; white/red bars) stimulated with LPS or S-protein (0.1 µg/ml) for 4 h and subsequent incubation with nigericin for 2 h. For statistical analysis, two-way ANOVA with Tukey post hoc test was used.

Data information: P-values for the DEGs were calculated using Wald test and P-adjusted values using the FDR/Benjamini-Hochberg approach. For the pathways and GO analysis, P-values were calculated by hypergeometric distribution and the adjusted P-value was calculated with the Benjamini–Hochberg method. Graphs show mean ± SEM. *P < 0.05; **P < 0.01.

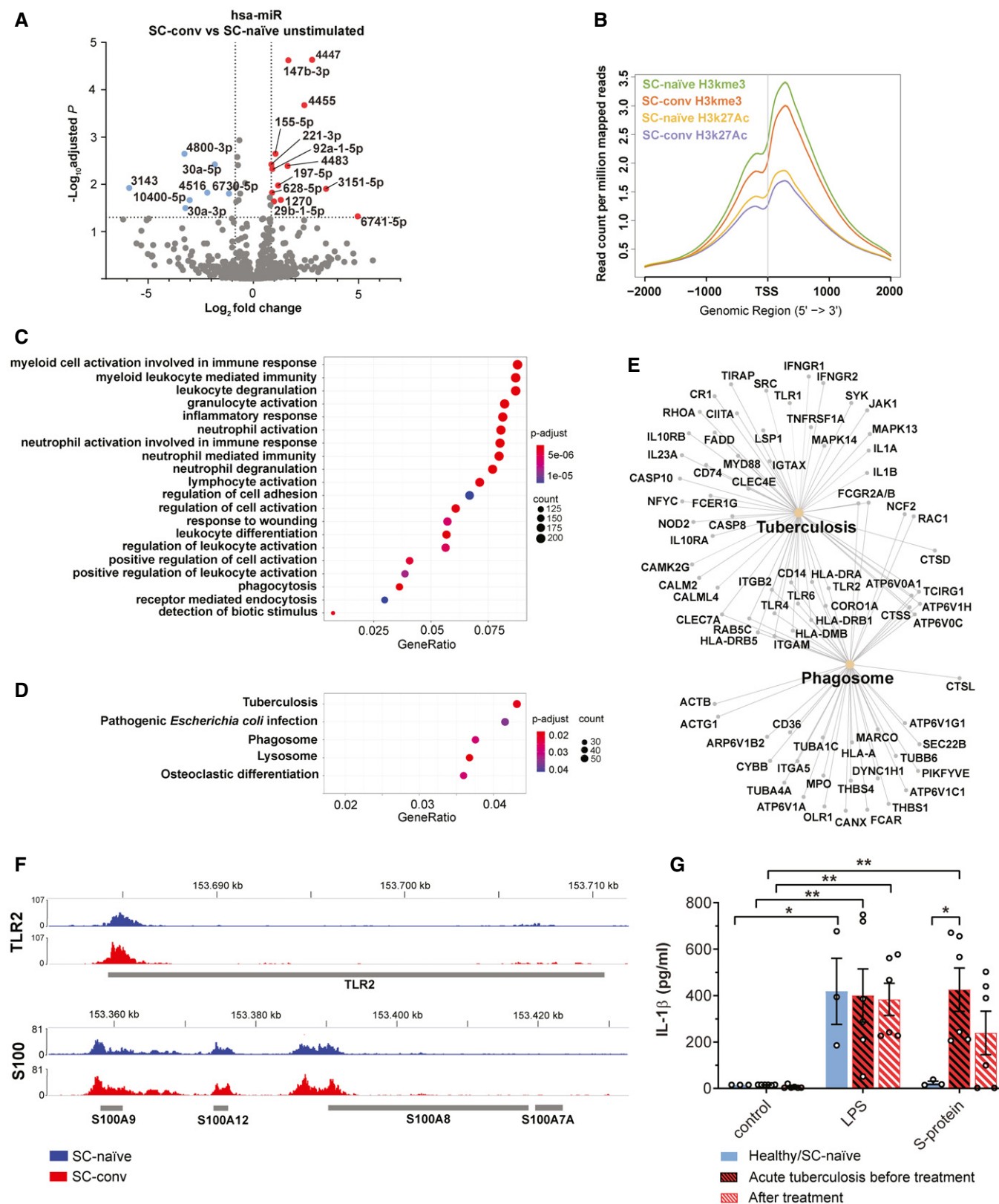

**Figure 6.**

inflammation-associated activation of myeloid cells and positive regulation of the immune response (Fig EV5). In addition, genes associated with the regulation of metal ion transport and interleukin-1 production were enriched in SC-conv samples (Fig EV5). Overall, these data on active promoters confirm findings made for H3K27ac enhancer marks in convalescent COVID-19 patient-derived monocytes. In sum, we provide evidence for inflammation and myeloid cell activation associated epigenetic histone modifications in COVID-19 convalescent patient-derived monocytes which were absent in cells derived from naïve individuals.

### S-protein-driven IL-1β secretion in tuberculosis patient-derived macrophages

To this point, our data suggest that the S-protein functions as a primer antigen selectively initiating cytokine release dependent on previous SARS-CoV-2 exposure. In contrast to an antigen- and pathogen-specific activation of the adaptive immune response, triggers of innate immune signaling are known to be non-specific. The transcriptomic signatures identified in S-protein stimulated and unstimulated COVID-19 patient-derived macrophages described above have also been detected in other infectious disease backgrounds or after exposure to pathogen-specific molecular patterns or vaccines. Upregulation of S100A8, S100A9, and S100A12, for example, has been linked to innate immune memory after vaccination with *Mycobacterium bovis*-BCG (Cirovic et al, 2020). A range of pathogens and vaccines trigger upregulation of the inflammation-associated microRNAs mir-155, 221, and 222 (Furci et al, 2013; Wang et al, 2014). We speculated that macrophages isolated from patients infected with COVID-19 non-related pathogens may also respond to S-protein stimulation. In tuberculosis, the innate immune response toward *Mycobacterium tuberculosis* (Mtb) largely relies on robust macrophage activation via TLR-2, NF-κB, and downstream inflammasome signaling (Gopalakrishnan & Salgame, 2016). Interestingly, our KEGG pathway analysis performed on epigenetically modified genes identified in CUT&RUN experiments revealed a signature which is associated with infectious diseases such as tuberculosis. Thus, we exposed macrophages from patients with active tuberculosis to S-protein and nigericin followed by quantification of IL-1β in the cell supernatant (Fig 6G, Appendix Table S1). Here, the S-protein led to a marked increase of IL-1β secretion similar to levels achieved by stimulation with LPS. This finding shows that the S-protein requires a certain degree of non-specific macrophage pre-activation for priming of the inflammasome. We also performed a longitudinal experiment with macrophages from tuberculosis patients that had received 6 months of antibiotic treatment. All patients were culture-confirmed Mtb-positive cases receiving an antibiotic treatment according to the resistance profile of the respective strains. In a proportion of patients, we observed a clear decline of IL-1β levels upon S-protein/nigericin stimulation of their macrophages (Fig 6G). Intriguingly, though, there was a subset of patients that remained reactive toward S-protein/nigericin stimulation, secreting relatively large amounts of IL-1β (Fig 6G). This clearly shows that immune memory of macrophages can be long-lasting and may exist for several months depending on the nature of the stimulatory agent.

## Discussion

Here we provide evidence for a SARS-CoV-2 structural component being a driver of pro-inflammatory cytokine secretion in human macrophages. Monocytes and macrophages play a key role in the SARS-CoV-2-induced inflammatory response (Merad & Martin, 2020). Our data indicate that the S-protein seems to have a dual role in both adaptive and innate immunity. Intriguingly, S-protein-driven inflammasome activation requires prior *in vivo* priming of monocytes since naïve individuals failed to secrete IL-1β when their macrophages were exposed to the S-protein *ex vivo* (Fig 7). Very little or no S-protein-driven inflammasome activation in cells derived from SARS-CoV-2 naïve individuals, as observed in our study, may be a surrogate for immune evasion and failing early viral control in host tissue, which is driven by the inflammasome and IL-1β as first lines of defense (Lapuente et al, 2018; Han et al, 2019; Huang et al, 2020). However, once patients are infected, macrophages become highly reactive, secreting large amounts of IL-1β upon stimulation. Here, the NLRP3 inflammasome may contribute to pathophysiology and exuberant inflammation as shown for ARDS, influenza A virus infection and for COVID-19 in postmortem lung biopsies (McAuley et al, 2013; Grailer et al, 2014; Rodrigues et al, 2021). However, it is important to note that in our hands, IL-1β levels secreted from patient-derived macrophages did not correlate with severity of disease or other factors like age or sex. Several patients could be identified with only mild symptoms and potent release of IL-1β upon stimulation of isolated macrophages. Furthermore, circulating monocytes converted to macrophages remained reactive in clinically healthy convalescent patients. Thus, with regard to macrophages driving inflammation in COVID-19, it may rather be a quantitative effect of large numbers of cells invading inflamed lung tissue that causes hyper-inflammatory syndromes and not a specific signature or hyper-reactive phenotype of monocytes and macrophages circulating in severely diseased individuals compared to those with mild disease or no symptoms at all. There is growing evidence from animal and human studies indicating that severity of disease correlates with the SARS-CoV-2 viral load in the lung, which may represent a potent trigger for macrophage invasion and activation (Imai et al, 2020; Westblade et al, 2020). In addition, local secretion of DAMPs such as S100 family proteins may spark further invasion of activated leukocytes and inflammation of lung tissue. Serum levels of the S100A8/S100A9 heterodimer (calprotectin) were recently shown to be elevated in severe COVID-19, and we can confirm significant upregulation of the respective genes in patient-derived macrophages well beyond viral clearance. One additional factor that may contribute to COVID-19 severity is the abundance of the SARS-CoV-2 binding receptor ACE2. However, we were not able to detect elevated levels of membrane-associated ACE2 on monocytes of diseased versus naïve individuals. In contrast, TLR2 was expressed to higher levels on patient-derived cells. Our RNA-seq and CUT&RUN data also revealed positive regulation of this receptor after SARS-CoV-2 infection. TLR2 stimulation was recently shown to be required for SARS-CoV-2 whole virus-induced pyroptosis (Rodrigues et al, 2021). However, it is unlikely that changes in TLR2 expression levels are exclusively responsible for differential activation of the inflammasome in S-protein exposed cells since other TLR2 agonists (zymosan) led to secretion of IL-1β in

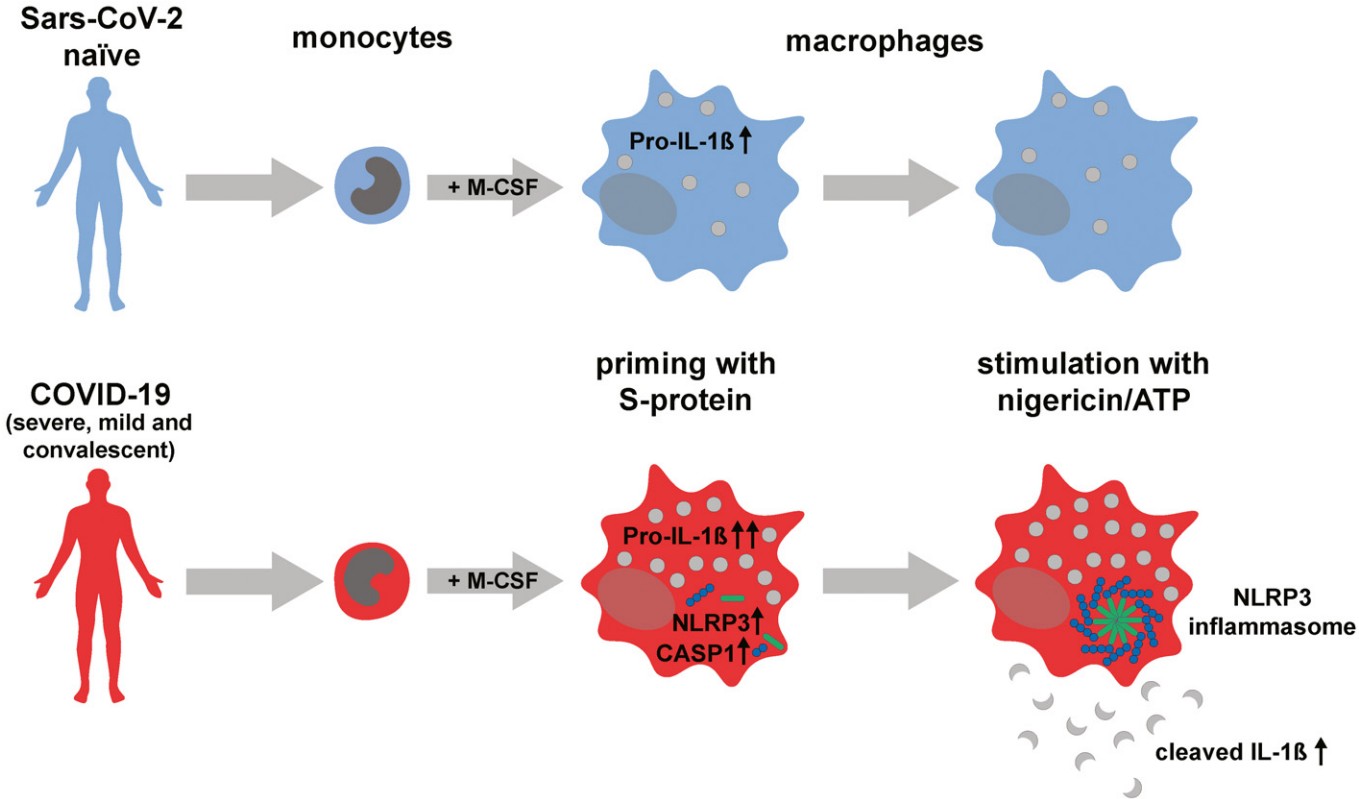

**Figure 7.** *In vivo* reprogramming of macrophages is required for SARS-CoV-2 S-protein dependent NLRP3 inflammasome formation and IL-1β secretion.

both COVID-19 patient-derived cells and cells from SARS-CoV-2 naïve controls.

COVID-19 is a novel pathogen currently affecting an immunologically naïve population thus providing an opportunity to gather a better understanding of timing and specificity of innate immune countermeasures of the host. In our *ex vivo* model, secretion of cleaved IL-1β seems to be extremely tightly controlled. Macrophages isolated from SARS-CoV-2 naïve individuals failed to activate the inflammasome even after exposure to very high concentrations of the S-protein and despite upregulation of IL-1β signaling-associated genes as shown by RNA-Seq and qRT–PCR. Some of the respective genes were expressed to significantly higher levels in COVID-19 patient-derived cells where exceeding of a certain threshold may lead to full activation of the inflammasome machinery and secretion of cleaved IL-1β. Interestingly, key components of this machinery such as NLRP3 and caspase-1 were exclusively expressed in COVID-19 patient-derived macrophages after priming with the S-protein. NEK7, a NLRP3 binding protein and essential component of the inflammasome was upregulated in COVID-19 patients. Upregulation of NEK7 has also been observed in other inflammatory diseases (Chen *et al*, 2019). In addition, our transcriptional analysis of macrophages from SARS-CoV-2 exposed or naïve individuals identified strong dysregulation of stress response-associated genes many of which have been linked to inflammasome activation indicating that, in patient-derived macrophages, the S-protein hits a specific intracellular environment that supports IL-1β cleavage and secretion. Intriguingly, this environment is not required for LPS or

zymosan-driven inflammasome processing which occurs to almost similar levels in macrophages from both SARS-CoV-2 naïve and convalescent individuals. Chronic exposure of the human innate immune system to LPS and zymosan derived from Gram-negative or fungal microbiota may necessitate non-selective inflammasome activation even in macrophages from naïve and healthy individuals. Thus, the macrophage inflammasome machinery is capable of discriminating between two antigen triggers resulting in differential activation depending on prior stimulation. At least with regard to the S-protein, this pre-activation step seems to be non-specific as shown by S-protein-driven IL-1β secretion in macrophages isolated from patients with active tuberculosis.

We also show that macrophage pre-activation is long-lasting and detectable in cells from convalescent patients several weeks after full recovery. Given the short half-life of circulating monocytes, there clearly is immunological memory or trained innate immunity in individuals having survived COVID-19 when responding to the S-protein. Previous data have shown that, after exposure to certain pathogens or vaccines, trained inflammatory monocytes are produced which become highly reactive following a second antigenic stimulus (Arts *et al*, 2018; O'Neill & Netea, 2020). However, a similar phenomenon has not been shown yet for a viral infection and a correlating PAMP. Innate immune memory is most likely generated through transcriptomic, epigenetic, and functional reprogramming of myeloid progenitors in the bone marrow and the respective marks can also be detected in blood monocytes (de Laval *et al*, 2020). We elucidate two possible mechanisms for long-lived

and selective reactivity of monocytes/macrophages from convalescent individuals. First, we were able to show that monocytes/macrophages from SARS-CoV-2 exposed individuals carry a highly distinct transcriptional signature associated with metal ion homeostasis and the cellular oxidation-reduction process all of which have been associated with modulation of innate immune functions. The dysregulated metal- and divalent-ion ($Zn^{2+}$, $Ca^{2+}$) binding proteins (metallothioneins and S100-familiy proteins) have emerged as modulators of innate immune functions and innate immune memory by regulating immune cell metal-ion homeostasis and redox balance (Ulas et al, 2017). Metallothioneins, which were strongly down-regulated in macrophages from COVID-19 patients, serve as antioxidants and modulate the activation of the transcription factor NF-κB. (Subramanian Vignesh & Deepe, 2017; Wang et al, 2018). Accordingly, treatment of macrophages from SARS-CoV-2 naïve individuals with reactive oxidant species (ROS) inducing agents resulted in responsiveness to S-protein/nigericin treatment. Notably, oxidative stress has recently emerged as an important modulator of NLRP3 inflammasome activation (Prochnicki & Latz, 2017).

Second, we identified an epigenetic- and microRNA-signature strongly associated with innate immune memory. Several of the 30 microRNAs, which were differentially regulated in convalescent patient-derived macrophages, are associated with long-lived reprogramming of innate immune functions. Myeloid cells expressing, for example, mir-155 are known to remain in a primed and hypersensitive state (Jablonski et al, 2016; Netea et al, 2016). Most importantly, we were able to show epigenetic histone modifications, which are associated with myeloid cell activation and the proinflammatory, IL-1β-driven host response. This signature was exclusively identified in monocytes from SARS-CoV-2 exposed convalescent individuals and not in naïve controls.

Thus, the specific signatures we observed in convalescent patient-derived monocytes and the corresponding intracellular environment seems to allow for long-lived and selective inflammasome activation in SARS-CoV-2 exposed individuals.

While we provide evidence for innate immune memory after SARS-CoV-2 exposure, the immunological phenomenon of non-specific innate immune cell priming is also being explored as a protective measure against COVID-19 and other viral diseases (Mantovani & Netea, 2020). Several clinical trials are investigating whether innate immune modulation induced by vaccination with the anti-tuberculous live vaccine BCG is effective in preventing severe forms of COVID-19 (Mantovani & Netea, 2020; O'Neill & Netea, 2020). Intriguingly, our in-depth transcriptomic, miRNA, and epigenetic analyses provide a certain degree of overlap in macrophage innate immune signaling identified after infection with Mycobacterium tuberculosis or BCG vaccination and SARS-CoV-2 exposure (Cirovic et al, 2020). Furthermore, we show that the SARS-CoV-2 S-protein is capable of priming inflammasome signaling in macrophages derived from tuberculosis patients again indicating common targets for the respective virus- or mycobacterium-derived PAMPs. Whether our ex vivo findings translate into protective effects of BCG vaccination requires further investigation.

To conclude, our findings contribute to an improved understanding of SARS-CoV-2 derived triggers of innate immunity pathways, which is required for rational designs of urgently needed therapeutic and preventive measures. In macrophages isolated from COVID-19 patients, inflammasome signaling is clearly pre-activated and can be primed with SARS-CoV-2-specific PAMPs making this pathway an attractive target against SARS-CoV-2-induced hyperinflammatory syndromes. In addition, innate immune activation and inflammasome formation is crucial for vaccine immunogenicity and for mounting an effective humoral immune response (Eisenbarth et al, 2008). Our data indicate that the primary SARS-CoV-2 vaccine antigen (S-protein) selectively and potently drives pro-inflammatory cytokine secretion in human monocytes. We provide comprehensive insights on the macrophage transcriptional landscape exposed to this important vaccine antigen before and after systemic exposure to SARS-CoV-2 which may help to predict safety and immunogenicity of currently applied vaccine strategies.

# Materials and Methods

### Patient samples and CD14[+] monocyte isolation

Blood samples were obtained from patients with proven SARS-CoV-2 infection at the University Hospital Cologne, Department I of Internal Medicine and from healthy donors. Infections were diagnosed by PCR from respiratory samples. For all human samples, written informed consent was obtained in accordance with the WMA declaration of Helsinki and the experiments conformed to the principles set out in the Department of Health and Human Services Belmont Report. The study was approved by the ethics committee of Cologne (identifier: 20–1,295). Detailed patients' information is provided in Appendix Table S1. PBMCs (peripheral blood mononuclear cells) were purified by density gradient centrifugation (Ficoll Plus, GE Healthcare, Chicago, IL, USA). CD14[+] cells were isolated from PBMCs by positive selection, using a monocyte isolation kit (Miltenyi Biotch, Bergisch Gladbach, Germany). $2.5 \times 10^5 / 5 \times 10^4$ CD14[+] cells were seeded into 24/96-well plates (TPP Techno Plastic Products AG, Trasadingen, Switzerland) and cultured for additional 5 days in Roswell Park Memorial Institute (RPMI) 1640 Medium (Thermo Fisher Scientific, Waltham, MA, USA) containing 10% fetal bovine serum (Thermo Fisher Scientific) and 50 ng/ml M-CSF (Miltenyi Biotec) for macrophage differentiation at 37°C and 5% $CO_2$.

### Cloning and expression of different spike protein variants

Ectodomain (MN908947; AA:1–1,207), RBD (MN908947; AA:331–524), and S2 (MN908947; AA:686–1,207) regions of the spike DNA were amplified from a synthetic gene plasmid (furin site mutated) using specific PCR primers. ORF8 PCR products were digested with the appropriate restriction enzymes and cloned into a modified sleeping beauty transposon expression vector containing for the ectodomain and S2-protein a C-terminal T4 fibritin foldon followed by a double Strep II purification tag. For recombinant protein production, stable HEK293 EBNA cell lines were generated employing the sleeping beauty transposon system (Kowarz et al, 2015). Briefly, expression constructs were transfected into the HEK293 EBNA cells, and after selection with puromycin, cells were expanded in triple flasks and protein production was induced by doxycycline. Supernatants of cells were harvested every 3 days, filtered and the recombinant proteins purified via Strep-Tactin®XT (IBA Lifescience, Göttingen, Germany) resin. Proteins were then

eluted by biotin containing TBS-buffer (IBA Lifescience, Göttingen, Germany), dialyzed against TBS-buffer, and aliquots stored at either 4°C or −80°C. All reusable materials were treated with 1 M NaOH for 2 h to remove any LPS contamination. Experiments were partially repeated with commercially available S-protein preparations which gave similar results (SARS-CoV-2 (2019-nCoV) Spike S1+S2 ECD-His Recombinant Protein, Sino Biological, Beijing, China).

### SARS-CoV-2 spike binding assay

For analysis of IgG interaction with SARS-CoV-2 protein, high binding 96-well ELISA plates (Corning Inc., Corning, NY, USA) were coated with SARS-CoV-2 spike protein (5 µg/ml) at 4°C overnight, washed 3× with PBS and blocked with PBS, containing 5% BSA (Carl Roth, Karlsruhe, Germany) for 60 min at RT. Thereafter, IgGs were tested at 3-fold dilutions (1:2) starting at concentrations of 166 µg/ml in PBS / 5% BSA for 120 min at RT. IgGs were isolated with Protein G Sepharose® 4 Fast Flow (GE Healthcare). The plates were washed 3× and incubated with horseradish peroxidase-conjugated goat anti-human IgG antibody (Jackson ImmunoResearch West Grove, PA, USA) (1:2500 in PBS/5% BSA) for 60 min at RT. ELISAs were developed using 2,2'-azino-bis(3-ethylbenzothiazoline-6-sulphonic acid) solution (ABTS, Thermo Fisher Scientific), and absorbance (OD 415–695 nm) was measured with absorbance reader (Tecan, Männedorf, Switzerland).

### Ex vivo stimulation of macrophages

Medium of differentiated macrophages was exchanged, and macrophages were incubated for 2 h at 37°C and 5% $CO_2$. After medium exchange, lipopolysaccharide (LPS) (5 µg/ml) (Sigma-Aldrich, St. Louis, MO, USA), SARS-CoV-2 proteins (0.1, 1 or 10 µg/ml) or Zymosan (10 µg/ml, Invivogen, Toulouse, France) (protein and concentrations for each experiment are stated in the figure legend) were added for additional 4 h in order to prime the inflammasome process. To activate IL-1β secretion, nigericin (5 µM) (Sigma-Aldrich) or ATP (5 mM) (Thermo Fisher Scientific) was added for 2 h at 37°C and 5% $CO_2$. For inhibitor experiments, cells were incubated prior first stimulation with either DMSO (Sigma-Aldrich), MCC950 (10 µM) (Sigma-Aldrich), VX-765 (5 µM) (Sigma-Aldrich), KINK-1 (5 µM) (Sigma-Aldrich), MMG-11 (25 µM) (Tocris, Bristol, UK), or blocking anti-TLR-2-IgA antibody (10 µg/ml, 100 and 1 ng/ml concentrations were used, Invivogen) for 2 h. For in vitro restimulation of healthy macrophages to induce oxidative stress, we pre-incubated the cells with FCCP (1 µg/ml, Sigma-Aldrich) and $H_2O_2$ (10 µg/ml, Millipore) for 24 h. Consequently, cells were stimulated as previously described. All assays were performed in technical duplicates. Supernatants were frozen down at −80°C for subsequent cytokine analysis.

### Quantification of IL-1β

IL-1β ELISA (BioLegend, San Diego, CA, USA and Thermo Fisher Scientific) was performed according to manufactures manual. Briefly, supernatant was diluted 1:50 in IL-1β ELISA Kit diluent and incubated for 2 h on previously coated 96-well ELISA plates (Thermo Fisher Scientific). All samples were measured in technical duplicates and concentration was determined with a corresponding standard curve provided by the IL-1β ELISA kit. OD was determined with Hidex Sense microplate reader (Hidex, Turku, Finland). Data were analyzed with Microsoft Excel (Microsoft, Redmond, WA, USA) and GraphPad Prism 8.0.2 software (GraphPad, San Diego, CA, USA).

### Immunoblot analysis

After isolation, macrophages were seeded in 24-well plated covered by 0.5 ml media. Differentiation and stimulation were performed as described above. Cell supernatant was precipitated with chloroform and methanol. Cell lysates were obtained, and the protein concentration was measured with the Pierce™ BCA Protein Assay Kit (Thermo Fisher Scientific). For Western blot analysis with supernatant, experiments were normalized to the number of input cells added to the respective wells in the experiment. Proteins or precipitated supernatant were subjected to SDS–PAGE (NuPAGE™ 4–12%, Bis-Tris, 10 well Gel, Thermo Fisher Scientific) and blotted on nitrocellulose membranes (Trans-Blot Turbo Transfer System, Bio-Rad Laboratories, Inc., Hercules, CA, USA). After blocking for 1 h, membranes were incubated with an primary antibody overnight at 4°C. The next day membranes were washed, incubated with a secondary antibody, and developed by chemiluminescent. Following antibodies were used: anti-cleaved IL-1β (1:1,000), anti-cleaved GSDMD (1:1,000), anti-pro-IL-1β (1:1,000), anti-β-actin (1:1,000), anti-NLRP3 (1:1,000), anti-p-NLRP3 (1:1,000), anti-ubiquitin (1:1,000), anti-NEK7 (1:1,000), anti-mouse IgG, HRP-linked Antibody (1:5,000), anti-rabbit IgG, HRP-linked Antibody (1:5,000) (all Cell Signaling Technology, Danvers, MA, USA). All immunoblot analysis were performed with at least $n = 3$ independent individuals from each experimental group, as indicated in the figure legend.

### Immunoprecipitation

Human macrophages were isolated as described before and seeded on 6-well plates with $2.5 \times 10^6$ cells per well. Stimulation was performed as described before. After stimulation, cells were washed two times with PBS, and afterward, 200 µl lysis buffer (1% Triton-X, 150 mM NaCl, 50 mM Tris, 1% SDS, 0.5% deoxycholate) was added. Cells were scraped and incubated for 20 min on ice. Lysates were centrifuged at $20,000 \times g$ for 20 min at 4°C. Afterward, lysates were diluted to 0.1% SDS with a dilution buffer (1% Triton-X, 150 mM NaCl, 50 mM Tris, 0.5% deoxycholate). Next, lysates were incubated for 1 h at 4°C on a rotating wheel with anti-NLRP3 antibody (1:200, Cell Signaling Technology). 50 µl protein-agarose beads (Santa Cruz) were added per probe and incubated overnight at 4°C on a rotating wheel. Finally beads were washed, proteins were eluted from the beads and immunoblot was performed as described before.

### Flow cytometry

Single-cell suspensions were prepared from PBMCs, and surface antigens were stained with fluorescently labeled antibodies: CD14 BV421 (M5E2) (BioLegend), CD16 APC-Cy7 (B73.1) (BioLegend), CD11c FITC (MJ4-27G12) (Miltenyi), HLA-DR PerCP (AC122) (Miltenyi), CD86 PE (IT2.2) (BioLegend), CD206 (19.2) (Becton Dickinson (BD), Franklin Lakes, NJ, USA), and CD282 FITC (TL2.1)

(BioLegend) (all in a 1:100 dilution). Cell death was quantified by using 5 µl 7-AAD /100 µl (BD). Data were acquired on a MACS-Quant 10 flow cytometer (Miltenyi). FlowJo (v10.6.2, FlowJo, LLC, Ashland, OR, USA) was used for data analysis and presentation.

### ACE2 detection

M-CSF differentiated macrophages (5 days) were used for detection of ACE-2 surface expression. Cells were detached using MACS buffer (Miltenyi Biotech) for 20 min at 37°C. Cells were blocked for 30 min using 10% FBS/PBS. For detection of surface ACE-2 expression, cells were incubated with anti-ACE-2 primary antibody (1:200, R&D systems) for 30 min at 4°C. Anti-goat-Alexa647 (Abcam) secondary antibody was used and incubated for 30 min at 4°C. Data were acquired on a MACSQuant 10 flow cytometer (Miltenyi). FlowJo (v10.6.2, FlowJo, LLC, Ashland, OR, USA) was used for data analysis and presentation.

### ASC-SPECK immunofluorescence staining

$8 \times 10^4$ CD14$^+$ monocytes were seeded per well into a 8-well chamber slide (Falcon) and differentiated with M-CSF as previously described. Upon stimulation as previously described, cells were washed 2-times with PBS and incubated with 4% PFA/PBS for 15 min at room temperature. Cells were additionally washed 2-times with PBS and consequently blocked for 1 h at room temperature with 5% FBS / 0.1% Tween-20 / 0.1% Triton-X in PBS. Anti-ASC-SPECK-Alexa488 (SantaCruz) antibody was added overnight at 4°C in 3% FBS / 0.1% Tween-20 / 0.1% Triton-X in PBS. Next, cells were washed 2-times with PBS and further incubated with DAPI (Thermo Fisher) for 10 min at room temperature. The slide was further washed 2-times with PBS and dried at room temperature. Finally, mounting media (Thermo Fisher) and coverslips were added. Pictures were acquired on a Olympus Fluoview FV1000 confocal microscope with 60× objective using the same microscope settings for all pictures. Pictures were analyzed using ImageJ and GraphPad Prism 8.0.2 software (GraphPad, San Diego, CA, USA).

### Cytokine detection assay

Cytokine quantification in EDTA-treated plasma and supernatants of macrophages stimulation experiments were performed with the Human Inflammatory Cytokine Kit from BD. Plasma samples and primary cell culture supernatants were diluted (1:2) with assay buffer and incubated with capture beads and PE detection reagent (all BD) for 1.5 h (plasma samples) or 3 h (supernatant samples) according to the manufacturer's instructions. Data were acquired on a MACSQuant 10 flow cytometer (Miltenyi) and analyzed with FlowJo (v10.6.2, FlowJo) (geometric mean fluorescence intensity (MFI) of each capture bead population). Cytokine concentrations were calculated by Microsoft Excel (Microsoft) with a standard curve of the MFI using provided cytokine standards.

### IL-1β expression analysis

For gene expression analysis, RNA from $1 \times 10^5$ macrophages was isolated using the RNeasy Mini Kit (Qiagen) in accordance with the manufacturer's instructions. Subsequently, cDNA was generated by reverse transcription with a Quantitect reverse transcription kit (Qiagen). Quantitative real-time PCR was used to measure expression levels of indicated genes. Samples were measured in technical triplicates in a 96-well plate Multicolor Real-Time PCR Detection System (IQ$^{TM}$5, Bio-Rad) using LightCycler®SYBR-Green I Mix (Roche, Basel, Switzerland). Data analysis was done based on linear regression of the logarithmic fluorescence values/cycle with the program LinRegPCR and target gene expression was normalized to the reference gene *actin*.

### Transcriptome Sequencing (RNA-Seq) and small RNA sequencing

CD14$^+$ cells were isolated as described before. $2.5 \times 10^5$ CD14$^+$ cells were seeded into 24-well plates (TPP Techno Plastic Products AG, Trasadingen, Switzerland) and cultured for additional 5 days in RPMI and 50 ng/ml M-CSF. After medium exchange, cells were stimulated with SARS-CoV-2 spike protein (1 µg/ml), LPS (5 µg/ml, LPS; Sigma-Aldrich), or left untreated. After 4 h, cells were washed, and total RNA was isolated with the mirVana$^{TM}$ Isolation Kit (Thermo Fisher Scientific) according to the manufacturer's instructions. RNA-Seq library prep was performed with 100 ng total RNA input and the NEBNext Ultra RNA library prep protocol (New England Biolabs, Ipswich, MA, USA) according to standard procedures. Libraries were validated and quantified (Tape Station 4200, Agilent Technologies, Santa Clara, CA, USA). Small RNA library preparation was done with 100 ng total RNA input/sample and the small RNA lib prep Kit (Lexogen, Wien, Austria) according to the standard protocol. Pooled libraries were size selected using 3% agarose cassettes (Biozym Scientific, Hessisch Oldendorf, Germany) and the BluePippin device (Sage Science, Beverly, MA, USA).

All libraries were quantified by using the KAPA Library Quantification Kit (VWR International, Radnor, PA, USA) and the 7900HT Sequence Detection System (Applied Biosystems, Foster City, CA, USA). Sequencing was done with NovaSeq6000 sequencers (Illumina, San Diego, CA, USA) with a PE100bp read length aiming at 50 M reads/sample (RNA-Seq) or a SR50bp read length aiming at 5 M reads/sample (small RNA). Demultiplexing and FastQ file generation was performed using Illumina's bcl2fastq2 software (v2.20.0).

The miRNA data were analyzed using the nf-core (Ewels *et al*, 2020) community-curated pipeline smrnaseq (v1.0.0). This pipeline performs adapter trimming with TrimGalore (v0.6.3) (http://www.bioinformatics.babraham.ac.uk/projects/trim_galore/) followed by alignment against human mature miRNAs and miRNA precursors (hairpins) (sequences obtained from miRBase version 22.1) (Kozomara *et al*, 2019) using bowtie (v1.2.2) (Langmead *et al*, 2009). The resulting BAM files were further processed using mirtop (http://mirtop.github.io) which generates a miRNA count table after summarizing unique isomiRs for each miRNA. The resulting count table was imported into R (v3.5.1), and the Bioconductor DESeq2 package (v1.20.0) (Love *et al*, 2014) was used for differential expression analysis according to the DESeq2 standard workflow.

RNA-seq data were analyzed using the Cologne Center for Genomics' in-house pipeline for RNA-Seq analysis based on the nextflow DSL (version 20.01.0 build 5264, (Di Tommaso *et al*, 2017)). In short, fastq files were adapter-trimmed with trimmomatic (v0.38) (Bolger *et al*, 2014), removing sequences of length < 18 nt after trimming, then mapped against the hg38 human reference genome and the gene assembly v101 using STAR aligner (v2.6.1a) with the

default settings. The count matrix was generated using subread (v1.6.4) (Liao Y, Smyth GK, Shi W, 2014) and deferentially expressed genes (DEGs) were called with DeSeq2 (v1.20.0) with a $Padj$. cutoff of 0.05. Overrepresented gene ontology biological processes and KEGG pathways were discovered using clusterprofiler (v3.17.0) (Yu *et al*, 2012) on up- or down-regulated DEGs (with $Padj < 0.05$ and a fold change of $> 2$ for either up or down-regulated). Volcano plots were generated with bioconductor package enchancedVolcano (v1.4.0) (https://github.com/kevinblighe/EnhancedVolcano) and heat maps with the heatmap.2 or pheatmap sub-function of gplots (v3.1.0 and v1.0.12, respectively) (https://github.com/talgalili/gplots and https://github.com/raivokolde/pheatmap, respectively).

## CUT&RUN

CUT&RUN was performed on 150K monocytes / sample adapting a previously described protocol (Skene & Henikoff, 2017). In brief, cells were washed twice with Wash Buffer (20 mM MHEPES, 150 mM NaCl, 0.5 mM Spermidine) and bound to activated Concanavalin A beads for 10 min at room temperature. Cell-bead suspension was then resuspended in antibody buffer (20 mM MHEPES, 150 mM NaCl, 0.5 mM Spermidine, 0.05% Digitonin, 2 mM EDTA) and incubated overnight at 4°C with H3K27ac (Active Motif #39133) and H3K4me3 (Active Motif #39159) antibodies (antibody dilution 1:50). Cell-bead suspension was then washed twice with Digitonin Buffer (20 mM MHEPES, 150 mM NaCl, 0.5 mM Spermidine, 0.05% Digitonin) and incubated with 2.5 µl EpiCypher CUTANA pAG-MNase, for 10 min. After washing samples twice with Digitonin buffer, 1 µl 100 mM $CaCl_2$ was added to samples which were incubated for 2 h at 4°C, rotating. To stop the reaction, STOP buffer (340 mM Nacl, 20 mm EDTA, 4 mM EGTA, 0.02% Digitonin, 50 µg/ml RNase A, 50 µg/ml Glycogen, yeast genomic DNA) was added in each tube. Samples were incubated at 37°C for 10 min at 500 rpm, and after centrifugation, liquid was collected, DNA was purified using Zymo DNA clean concentrator kit, and DNA was eluted in elution buffer. For library preparation, we used the TruSeq DNA nano kit and protocol from Illumina with 15 cycles of PCR. Libraries were validated and quantified (Agilent Tape Station 4200). A pool of all libraries was quantified by using the Peqlab KAPA Library Quantification Kit and the Applied Biosystems 7900HT Sequence Detection System. Sequencing was done with NovaSeq sequencers (Illumina) and a PE100bp read length aiming at 10 M reads/sample.

## CUT&RUN data analysis

Raw fastq files were mapped and treated as described by Zhu *et al* (PMID: 31500663). The GRCh38 human reference genome was used to analyze peaks. We used the Genrich peak caller that calls peaks for multiple replicates collectively. This software first analyses the replicates separately, with $P$-values calculated for each. At each genomic position, the multiple replicates' $P$-values are then combined by Fisher's method. The combined $P$-values are converted to q-values, and peaks are called. In order to assess the differential coverage of binding sites, we used the THOR software as described in (Allhoff *et al*, 2016). KEGG and GO analyses of CUT&RUN data were performed as described above. Differentially enriched loci of H3K27ac peaks associated with promoter sites $\pm$ 3,000 bps were used.

### The paper explained

**Problem**

SARS-CoV-2 causes a pandemic with more than 100 million infected people so far. SARS-CoV-2 infection can result in coronavirus disease 2019 (COVID-19) which leads to massive systemic and pulmonary inflammation in a subset of patients. The structural components of SARS-CoV-2 and the exact triggers leading to inflammation are not well described.

**Results**

The NLRP3 inflammasome is a multiprotein complex required for secretion of the pro-inflammatory cytokine IL-1β, which plays a key role in COVID-19 hyperinflammatory syndromes. The SARS-CoV-2 S-protein, a surface exposed viral receptor binding protein and important as vaccine antigen triggers NLRP3 inflammasome activation and cytokine secretion selectively in COVID-19 patient-derived macrophages. SARS-CoV-2 infection leads to reprogramming of human macrophages providing an intracellular landscape that allows for rapid inflammasome assembly.

**Impact**

Our study provides a rational for pharmacologically targeting the NLRP3 inflammasome and associated cytokines in severe cases of COVID-19. Profound alteration of macrophage gene activation and expression for several weeks to months after infection in both severe and mild COVID-19 patients can provide a better understanding of post-COVID inflammatory syndromes. Our data will help to evaluate immunogenicity of S-protein-based vaccine constructs with regard to stimulation of innate immune signaling.

## Statistical analysis

Statistical analysis was performed with GraphPad Prism 8.0.2 software (GraphPad, San Diego, CA, USA). Statistical parameters (value of n, statistical calculation, etc.) are stated in the figure legend. $P$-values less than or equal to 0.05 were considered statistically significant. Exact $P$-values of figures are listed in Appendix Table S2.

## Data availability

Transcriptome data have been deposited in the Gene Expression Omnibus (GEO; https://www.ncbi.nlm.nih.gov/geo/) of the National Center for Biotechnology Information under the accession number GSE173488: https://www.ncbi.nlm.nih.gov/geo/query/acc.cgi?acc=GSE173488. CUT&RUN data have been deposited under the accession number GSE173568: https://www.ncbi.nlm.nih.gov/geo/query/acc.cgi?acc=GSE173568. All other data that support the findings of this study are available from the corresponding author on request.

**Expanded View** for this article is available online.

## Acknowledgements

We are grateful to patients and healthy individuals for donating blood used in this investigation. We gratefully thank Jason McLellen, Daniel Wrapp, and Nianshuang Wang for sharing the prefusion-stabilized S-protein plasmid. We are grateful to Edeltraut van Gumpel and Ute Sandaradura de Silva for technical support. Swati Parekh (MPI-Age) was of great help during the CUT&RUN experiments. J. R. receives funding from the Thematic Translational Unit Tuberculosis (TTU TB, Grant number TTU 02.806 and 02.905) of the German Center

of Infection Research (DZIF). Financial support was also received from the German Research Foundation (DFG RY 159/3-1, SFB1403 (project number 414786233)) and the Center for Molecular Medicine Cologne (ZMMK-CAP8). Funding was received from the COVIM project of NaFoUniMedCovid19 network (FKZ: 01KX2021) and the newERA4TB IMI network. A.S. is supported by the Cologne Clinician Scientist Program (CCSP), funded by the German Research Council (FI 773/15-1) and receives funding from the Cologne Fortune Program/Faculty of Medicine, University of Cologne. S.J.T receives funding from the Cologne Fortune Program/Faculty of Medicine, University of Cologne. H.S.E. is supported by a fellowship from the Cologne Fortune Program/Faculty of Medicine, University of Cologne. J.F. received funding from the German Center for Infection Research (DZIF) (TI 07.005_Fischer_00), the Cologne Fortune Program, and the medical faculty of the University of Cologne, Germany (Gusyk funding). I. S. receives funding by the German Center for Infection Research (DZIF) (Grant number TTU 02.806 and 02.905, Grant number TI 07.001_SUAREZ_00). P.K. has received non-financial scientific grants from the Cologne Excellence Cluster on Cellular Stress Responses in Aging-Associated Diseases, University of Cologne, Cologne, Germany. M.K. is supported by the German Research Foundation (DFG FOR2722). P.N., J.A., T.G., and S.M. are supported by the German Research Foundation (West German Genome Center INST 216/981-1 to the University of Cologne). O.A.C is supported by the German Federal Ministry of Research and Education, is funded by the German Research Foundation under Germany's Excellence Strategy—CECAD, EXC 2030—390661388. P.T. received funding from the Max Planck Society. A.P. is supported by Onassis Foundation (Scholarship ID: 047-1/2019-2020). Open Access funding enabled and organized by Projekt DEAL.

## Author contributions

SJT and AS contributed samples, planned and performed experiments, analyzed data, and wrote the manuscript; TG, SM, FE, FD, and PT analyzed RNA-seq/miRNA dataset. CK and MZ performed experiments and analyzed data; HSE, JF, M-CA, JC, KV, JG, SW, US, and AP performed experiments, OAC, BB, PK, JJM, HG, IS, MH, GF, NJ, HK, and OAC were involved in clinical care of patients, provided biosamples, and discussed data, CL, JA, PN, HK, and FK planned experiments, analyzed, and discussed data; MK provided SARS-CoV-2 proteins and performed experiments, JR directed the study and wrote the manuscript.

## Conflict of interest

The authors declare that they have no conflict of interest.

## For more information

i    https://www.who.int/emergencies/diseases/novel-coronavirus-2019/
ii   https://www.covid19treatmentguidelines.nih.gov/

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
