## [Review Process File · EMBO Molecular Medicine]

Long-lived macrophage reprogramming drives spike protein-mediated inflammasome activation in COVID-19

Sebastian Theobald, Alexander Simonis, Theodore Georgomanolis, Christoph Kreer, Matthias Zehner, Hannah Eisfeld, Marie-Christine Albert, Jason Cchen, Susanne Motameny, Florian Erger, Julia Fischer, Jakob Malin, Jessica Gräb, Sandra Winter, Andromachi Pouikli, Friederike David, Boris Böll, Philipp Koehler, Kanika Vanshylla, Henning Gruell, Isabelle Suarez, Michael Hallek, Gerd Fätkenheuer, Norma Jung, Oliver Cornely, Clara Lehmann, Peter Tessarz, Janine Altmüller, Peter Nuernberg, Hamid Kashkar, Florian Klein, Manuel Koch, and Jan Rybniker

DOI: [10.15252/emmm.202114150](https://doi.org/10.15252/emmm.202114150)

Corresponding author: Jan Rybniker (jan.rybniker@uk-koeln.de)

Review Timeline:

Submission Date:	3rd Dec 20
Editorial Decision:	24th Dec 20
Revision Received:	17th Feb 21
Editorial Decision:	8th Mar 21
Revision Received:	29th Apr 21
Editorial Decision:	20th May 21
Revision Received:	26th May 21
Accepted:	28th May 21

Editor: Zeljko Durdevic

Transaction Report:

Dear Dr. Rybniker,

Thank you for the submission of your manuscript to EMBO Molecular Medicine. We have now received feedback from the three reviewers who agreed to evaluate your manuscript.

As you will see from their reports pasted below, the referees acknowledge the interest of the study, however they also have serious and partially overlapping concerns that preclude further consideration of the article at this time. Given the nature of these criticisms, addressing all the referees' comments would require a lot of additional work, time and effort. As clear and conclusive insight into a novel clinically relevant observation is key for publication in EMBO Molecular Medicine, I am afraid that we do not feel it would be productive to call for a revised version of your manuscript at this stage and therefore we cannot offer to publish it.

Given the potential interest of the findings, we would, however, be willing to consider a new manuscript on the same topic if at some time in the near future you obtained data that would considerably strengthen the message of the study and address the referees concerns in full. To be completely clear, however, I would like to stress that if you were to send a new manuscript this would be treated as a new submission rather than a revision and would be reviewed afresh, in particular with respect to the literature and the novelty of your findings at the time of resubmission. If you decide to follow this route, please make sure you nevertheless upload a letter of response to the referees' comments.

I am sorry that I could not bring better news this time and hope that the referee comments are helpful in your continued work in this area.

Yours sincerely,

Zeljko Durdevic

***** Reviewer's comments *****

Referee #1 (Remarks for Author):

General Comments: This MS describes the ability of recombinant SARS-CoV-2 spike protein (S-protein) to act as an ex vivo "signal 1" priming stimulus for NLRP3 inflammasome assembly in human macrophages differentiated from the blood monocytes of COVID-19 patients but not virus naive "control" donors. This suggests a model of trained innate immunity whereby myeloid progenitors are epigenetically reprogrammed during in vivo S-protein exposure to stably upregulate sensors of S-protein. The MS presents functional analyses and transcriptomic screens which support elevated expression of TLR2 as a potential S-protein receptor that can drive NFkB-dependent induction of NLRP3 and IL1-b as "primed" components of the NLRP3 inflammasome signaling axis in monocyte/macrophages from COVID-19 patients. As a consequence, exposure of these primed macrophages to canonical "signal 2" stimuli which disrupt ion homeostasis (e.g., nigericin or ATP-activated P2X7 receptor channels) facilitates rapid assembly of NLRP3/caspase-1 inflammasome with consequent proteolytic maturation and release of IL-1b, a known component of the inflammatory "cytokine storm" in COVID-19 patients. Other data indicate that: 1) "trained" monocyte/macrophages from convalescent COVID-19 subjects are also defined by altered expression of various miRNA's associated with enhanced proinflammatory tone; and 2) monocytes/macrophages isolated from MTB patients are defined by a similar trained innate response to an ex vivo S-protein priming of the NLRP3 inflammasome signaling axis.

These are timely, relevant and mechanistically provocative findings. However, some additional analyses are required to extend and unequivocally support the major conclusions, particularly the presumptive role of TLR2 as a target for S-protein.

Specific Comments:

1. Figure 1D: Given the conclusion that unregulated TLR2 is a likely receptor mediator of the S-protein dependent inflammasome priming, the authors need to include a canonical TLR2 stimulus (e.g. Pam3CSK) in addition to LPS, as a highly relevant signal 1 priming agent for comparison with S-protein.
- 2) Figure 1D: There seems to be increased LPS priming response in the COVID-19 derived macrophages but statistical analysis is neither indicated nor discussed.
- 3) Figure 2B: Western blot analysis should include proIL-1b. It should be straightforward to reprobe these same blots with anti-IL1b.
- 4) Figure 2D: Western blot analysis of the supernatants should include another normally cytosolic protein (e.g. GAPDH) as further support (beyond Supplemental Fig 2E) of inflammasome-driven pyroptosis in the S-protein primed macrophages from COVID-19 subjects ,but not virus-naive controls.

- 5) Figure 3A (minor): the legend should specify that the cells were nigericin-stimulated post S-protein priming.
- 6) Figure 3F (minor): the legend indicates n=10 for convalescent subject derived samples. Presumably, there were multiple analyses on samples from the 4 subject listed in Supplemental Table 1C. This should be clarified in the text and legend.
- 7) Figures 3F/G/H: Similar to the point raised in comment 1, these analyses with convalescent macrophages should include priming with a canonical TLR2 agonist (e.g., Pam3CSK) for comparison with the actions of S-protein and LPS.
- 8) Figure 5B: Similar to the point raised in comments 1 and 7, these analyses with macrophages from MTB subjects should include priming with a canonical TLR2 agonist (e.g., Pam3CSK) for comparison with the actions of S-protein and LPS.
- 9) The authors don't indicate whether their analyses included evaluation of ACE2 expression in the trained monocytes/ macrophages from the COVID-19 subjects and the MTB subjects. This should be included..

Referee #2 (Comments on Novelty/Model System for Author):

The technical quality cannot be judged because the methods are not incorporated in any of the files to review. The methods are particularly important in this case due to some discrepancies with the published literature. Authors need to make an statement on power calculation for this study.

There is very little doubt of the interest of the work. The direct clinical impact it is a bit limited since we already know the activation of NLRP3 in COVID19 patients, and the correlation of IL1 and severity. In contrast, the findings showing long live re programming are intriguing and most likely will warrant future research.

The research model (PBMCS) is state-of-the-art. The reagents (inhibitors) are validated in the inflammasome field interrogating human systems.

Referee #2 (Remarks for Author):

This manuscript by Theobald and colleagues shows that SARS-CoV-2 S protein can act as a signal 1 in TLR2-dependent manner but surprisingly only in macrophages derived from COVID19 patients. Colleagues report data on gene expression signature of monocytes hinting towards a long-live re programming. However, no experimental evidence is provided to support this hypothesis.

Comments.

1. The authors are presenting somewhat misleading information based on their results. Upon reading the text a non-expert may get the impression that the S protein triggers NLRP3 activation, which is not the case since it only acts as a priming signal. I urge the authors to reformulate the text and clearly indicate that the S protein acts only as a priming signal.
2. Interrogating naive PBMCs, it will be relevant to test whether S protein also result in AIM2 activation following addition of DNA. In other words to test whether the phenotype observed only affects only NLRP3 or it is a general inflammasome issue. This simple experiment will reinforce the conclusion of S protein as signal 1.

3. It is remarkable that despite activating NF- κ B signalling, transcription of different cytokines and also NLRP3 and AIM2, authors cannot find NLRP3 activation in naive cells upon challenge with the S protein. This is perhaps one of the most surprising findings of the manuscript. Authors need to confirm by immunoblotting the presence of pre IL1 in the lysates of challenged cells (time course experiment). Authors need to show that there are no differences on the interaction of the S protein between different cells. Another experiment to be included is the secretion of TNF following stimulation (in naive and COVID19 cells if the latter is possible).
4. The patients' characteristics need to be presented in more detail: treatments received, number of days at the hospital and whether they have received any antibiotic treatment.
5. Ferritin is known to activate inflammasome. This might be a coconfounding factor in the experiments, and may have a long term impact on macrophage reprogramming. This should be discussed as a limitation.
6. It has been recently published that SARS-CoV-2 induces inflammasome activation in PBMCs (NLRP3 as well as potentially other unknown one(s) *J Exp Med* (2021) 218 (3): e20201707) from naive individuals. This finding using whole virus contradicts the results of this manuscript. This should be discussed as a significant limitation and it should be emphasized the relevance of using the whole pathogen versus purified products.
7. Authors need to discuss the biological relevance of the system employed: (i) is there any case in vivo in which cells will be challenged with the S protein (extracellularly) but not in the context of the viral particle? (ii) are the levels tested physiologically relevant? This information is mostly likely unknown and, therefore, this should be considered as a limitation.
8. This reviewer does not consider that the tuberculosis data add too much to this work. Does this too imply that the cells need to be activated to sense the S protein and therefore respond?
9. The miRNA is reported as a catalogue of genes but it is not mechanistically integrated in the work.
10. To sustain the statement of long-term reprogramming significant additional evidence needs to be included, including histone analysis and genome-wide CHIP data. This reviewer concurs with the impression of the authors; however in the absence of hard-core data authors need to tone down significantly their statement (including changing the title) and keep this as a hypothesis to be validated with further research.

Referee #3 (Remarks for Author):

The manuscript by Theobald et al. studies the response of Covid-naïve versus Covid-exposed monocyte-derived macrophages to the Spike protein of SARS CoV-2. The study finds distinct gene expression signatures between Covid-naïve and Covid-exposed populations and argues that this shows an innate immune memory that manifests in exacerbated inflammatory responses - mostly centered on IL-1 β . The studies are, of course, timely and the available data is important to mine. Unfortunately, the manuscript doesn't solidify its main findings and the result is a series of poorly connected experiments with alternative explanations never ruled out. Enthusiasm for the manuscript in its present form is therefore muted. Major and Minor critiques are as follows:

Major:

1. The comparison of the Spike protein priming the inflammasome is made to LPS, a TLR4 activator. Despite this, the authors show increased expression of TLR2 and then later show that a TLR2 inhibitor blunts the effect of the Spike protein on inflammasome priming. If TLR2 is sensing the Spike protein, shouldn't a TLR2 agonist be the true comparison here, not LPS?
2. Along these same lines, the role of TLR2 is hinted at, but never established. Do TLR2-null cells respond to the S protein? Much of the manuscript suffers from this problem - an interesting result is hinted at, but never solidified.
3. Inflammasome activation is never actually shown - Only IL-1b release. Does ASC oligomerize? Is Gasdermin D cleaved? Is Caspase-1 activated?
4. There is bimodal distribution in the response to the Spike protein. This can be seen in Figure 1D (last bar - bar 10) and in Figure 2A - DMSO treated. From the data shown in Figure 2G, those two patients studied must have been the highest responders in 1D and 2G. Given this, it isn't clear that patients generally respond as in 2G. This again is an interesting finding that isn't solidified.
5. The gene expression data is parsed in a number of ways, but the in vivo effect of the changes in gene expression isn't shown. For instance in Figure 3C, it shows that MAPK signaling is severely altered, but this is never studied. These sorts of experiments are hypothesis-generating, but become important when those hypotheses are tested. Does MAPK signaling affect the Covid inflammatory response? Almost every pathway shown could be studied. Instead, the data is presented but not in the context of a follow up experiment that shows its role in Covid inflammation. The same critique could be made of Figure 4.
6. The argument is that Covid induces a sort of innate immune memory. Mechanistically, for reasons that are unclear, this is attributed to miR-RNA expression (Figure 5). This is never experimentally tested however. It isn't unexpected that miRs change upon Covid exposure - that doesn't prove miRs are causing this though. This needs to be shown. As an aside, I would think an epigenetic phenomenon would be more likely.
7. The Tb data is thrown into the manuscript without much context. Patients with acute TB respond to the S protein, but in context of the manuscript is the question of whether recovered Tb patients respond to the S protein. What about other infections like influenza or patients with chronic inflammatory disease? Again, this interesting finding is never verified or studied in depth.

Minor

1. Pro-IL1b needs to be shown in Western blots (esp 2D). This is especially true as Figure 1C implies that Covid-exposed macrophages have higher basal levels of pro-IL1b and might be primed at baseline - not only at S protein exposure.

2. It isn't clear how Western blots are quantified. They are studying supernatant but quantifying it to an area on the gel with no signal? It needs to be quantified to something.

3. The figure presentation in Figure 2 is confusing. The box explaining the colors doesn't apply to every panel and its very distracting. Each panel should be outlining what's being studied.

4. Why is the IL-1b released in Figure 3H so much lower in Figures 1D or 2F?

5. The authors note the inflammasome-related expression of Pycard and Caspase-8 mRNA is significantly down in Figure 3D. Shouldn't this blunt the inflammasome (especially pycard (ASC)).

In summary, while the manuscript is timely and there are some interesting findings, none of these findings is studied in sufficient depth to be convincing. This fact, coupled with numerous experimental issues, significantly diminishes enthusiasm for the manuscript.

Rebuttal for Theobald et al., "Long-lived macrophage reprogramming drives spike protein-mediated inflammasome activation in COVID-19"

***** Reviewer's comments *****

Referee #1 (Remarks for Author):

General Comments: This MS describes the ability of recombinant SARS-CoV-2 spike protein (S-protein) to act as an ex vivo "signal 1" priming stimulus for NLRP3 inflammasome assembly in human macrophages differentiated from the blood monocytes of COVID-19 patients but not virus naive "control" donors. This suggests a model of trained innate immunity whereby myeloid progenitors are epigenetically reprogrammed during in vivo S-protein exposure to stably upregulate sensors of S-protein. The MS presents functional analyses and transcriptomic screens which support elevated expression of TLR2 as a potential S-protein receptor that can drive NFkB-dependent induction of NLRP3 and IL1-b as "primed" components of the NLRP3 inflammasome signaling axis in monocyte/macrophages from COVID-19 patients. As a consequence, exposure of these primed macrophages to canonical "signal 2" stimuli which disrupt ion homeostasis (e.g., nigericin or ATP-activated P2X7 receptor channels) facilitates rapid assembly of NLRP3/caspase-1 inflammasome with consequent proteolytic maturation and release of IL-1b, a known component of the inflammatory "cytokine storm" in COVID-19 patients. Other data indicate that: 1) "trained" monocyte/macrophages from convalescent COVID-19 subjects are also defined by altered expression of various miRNA's associated with enhanced proinflammatory tone; and 2) monocytes/macrophages isolated from MTB patients are defined by a similar trained innate response to an ex vivo S-protein priming of the NLRP3 inflammasome signaling axis.

These are timely, relevant and mechanistically provocative findings. However, some additional analyses are required to extend and unequivocally support the major conclusions, particularly the presumptive role of TLR2 as a target for S-protein.

Specific Comments:

1. Figure 1D: Given the conclusion that unregulated TLR2 is a likely receptor mediator of the S-protein dependent inflammasome priming, the authors need to include a canonical TLR2 stimulus (e.g. Pam3CSK) in addition to LPS, as a highly relevant signal 1 priming agent for comparison with S-protein.

We would first like to thank reviewer 1 for evaluating our manuscript and for providing important points of critique that we used to improve this fully revised version.

We agree that TLR2 binding PAMPs may represent an interesting control for our observations and added data generated with zymosan, a well described TLR2 agonist. Pilot experiments were performed with both Pam3CSK and Zymosan, the latter clearly gave better signals in our assays.

Zymosan was also functional in both macrophages from naïve donors and in macrophages from patients. This stands in contrast to what we observed with S-protein stimulation. Nevertheless, zymosan treated patient macrophages gave higher signals than naïve ones. Overall, the signal obtained from zymosan is comparable to what we see in LPS treated cells. We also performed TLR2 blocking experiments using monoclonal antibodies. These experiments also confirm involvement of TLR2 in S-protein signaling. However, changes in TLR2 expression levels do not fully explain differences in inflammasome activation observed in macrophages derived from different donors. In the revised manuscript, we provide additional explanations: altered REDOX states seem to contribute to inflammasome activation in COVID-19 patient

derived cells (Fig. 5E). In addition, a novel dataset on epigenetic modification of monocytes from patients (CUT&RUN experiments) give insight on chromatin changes associated with the pro-inflammatory response in these cells (Fig. 6).

New data on zymosan are presented in Fig. 4D and E.

2) Figure 1D: There seems to be increased LPS priming response in the COVID-19 derived macrophages but statistical analysis is neither indicated nor discussed.

In Fig. 1D there is indeed a significant increase of IL-1 β secretion in COVID-19 patients compared to healthy and naïve controls. This is primarily due to acutely diseased patients whose macrophages seem to be activated in a way that leads to somewhat higher LPS-levels. To correct for this effect, we did many experiments with convalescent patients who still reacted to the S-protein and LPS. Here, LPS reactivity seems to be stable over time while S-protein reactivity declines strongly. More importantly, we never observed a total lack of cleaved IL-1 β in supernatants of LPS treated cells (both naïve and diseased). Nevertheless, we mention the effect observed for LPS in Fig. 1D and highlight the statistically significant difference we observed.

The section now reads: "LPS and nigericin treatment led to higher IL-1 β secretion in macrophages from COVID-19 patients. However, macrophages from naïve controls clearly secreted significant amounts of IL-1 β which was not the case for S-protein stimulated naïve cells."

3) Figure 2B: Western blot analysis should include proIL-1 β . It should be straightforward to reprobe these same blots with anti-IL1 β ."

We understand that this is an important experiment. We performed the requested immunoblot which correlates well with the qRT-PCR data for IL-1 β . The data obtained confirm our qRT PCR experiments on the same target (New Fig. 2F).

4) Figure 2D: Western blot analysis of the supernatants should include another normally cytosolic protein (e.g. GAPDH) as further support (beyond Supplemental Fig 2E) of inflammasome-driven pyroptosis in the S-protein primed macrophages from COVID-19 subjects, but not virus-naïve controls.

The requested GAPDH quantification via immunoblot from supernatants has been performed. It clearly shows leakage of this cytosolic protein in LPS/nigericin treated cells (see below). As expected, the protein is only detected in supernatants of COVID-19 patient derived macrophages treated with the S-protein and nigericin.

In addition, we also provide a new immunoblot showing cleavage of GSDMD upon stimulation with the S-protein/nigericin (Fig. 2H). GSDMD-N forms the pores required for pyroptotic cell death. It is a substrate

of caspase 1. In summary we provide several lines of evidence for S-protein driven inflammasome formation leading to pyroptotic cell death.

Since GSDMD cleavage is the hallmark of pyroptotic cell death, we leave it to the discretion of reviewer 1 and the editor whether we should also include the GAPDH immunoblot. In the current version, the blot was not included.

5) Figure 3A (minor): the legend should specify that the cells were nigericin-stimulated post S-protein priming.

This has been changed. We thank reviewer 1 for this advice.

6) Figure 3F (minor): the legend indicates n=10 for convalescent subject derived samples. Presumably, there were multiple analyses on samples from the 4 subject listed in Supplemental Table 1C. This should be clarified in the text and legend.

We thank reviewer 1 for highlighting this lack of clarity in our manuscript.

The four subjects we enrolled for S1C were newly enrolled individuals since we matched them for RNA-Seq analysis with naïve controls. The clinical data on convalescent subjects used for other experiments are listed in Table 1B, this is now mentioned in the respective table.

In the section correlating with S1C we now clarify that we included “additional” individuals for this analysis.

7) Figures 3F/G/H: Similar to the point raised in comment 1, these analyses with convalescent macrophages should include priming with a canonical TLR2 agonist (e.g., Pam3CSK) for comparison with the actions of S-protein and LPS.

These data were repeated with zymosan as described above (Fig. 4D).

8) Figure 5B: Similar to the point raised in comments 1 and 7, these analyses with macrophages from MTB subjects should include priming with a canonical TLR2 agonist (e.g., Pam3CSK) for comparison with the actions of S-protein and LPS.

We repeated the experiment with treated tuberculosis patients since we were using our biobanked TB patient cohort macrophages for experiments requested by reviewer 3. In these patients after 6 month of treatment we did see, as expected, a signal from zymosan. The signal is comparable to the LPS signal and the signal from healthy controls. Unfortunately, we did not have enough monocytes in our biobank to do the experiment with untreated TB patients. However, since we did get a potent signal from healthy controls and treated patients, we strongly believe that the results would be similar in non-treated patients also. The data are shown below. We would prefer not to add this dataset to the manuscript since this would in our view confuse the reader.

9) The authors don't indicate whether their analyses included evaluation of ACE2 expression in the trained monocytes/ macrophages from the COVID-19 subjects and the MTB subjects. This should be included.

We performed FACS analysis on macrophages derived from healthy controls and COVID-19 patients. There was no difference in the amount of ACE2 detected (Supplementary Fig. 2I).

Referee #2 (Comments on Novelty/Model System for Author):

The technical quality cannot be judged because the methods are not incorporated in any of the files to review. The methods are particularly important in this case due to some discrepancies with the published literature. Authors need to make a statement on power calculation for this study. There is very little doubt of the interest of the work. The direct clinical impact it is a bit limited since we already know the activation of NLRP3 in COVID19 patients, and the correlation of Il1 and severity. In contrast, the findings showing long live re programming are intriguing and most likely will warrant future research.

The research model (PBMCS) is state-of-the-art. The reagents (inhibitors) are validated in the inflammasome field interrogating human systems.

We thank reviewer 2 for taking the time to evaluate our manuscript. We regret that for an unknown reason reviewer 2 was not able to evaluate the extensive methods section we had uploaded together with our manuscript which was part of the main manuscript file. We had also uploaded a large "reagents table" as an additional file. We hope that Reviewer 2 will have a chance to evaluate the methods section in our fully revised version of the manuscript.

Referee #2 (Remarks for Author):

This manuscript by Theobald and colleagues shows that SARS-CoV-2 S protein can act as a signal 1 in TLR2-dependent manner but surprisingly only in macrophages derived from COVID19 patients. Colleagues report data on gene expression signature of monocytes hinting towards a long-live re programming. However, no experimental evidence is provided to support this hypothesis.

Comments.

1. The authors are presenting somewhat misleading information based on their results. Upon reading the

text a non-expert may get the impression that the S protein triggers NLRP3 activation, which is not the case since it only acts as a priming signal. I urge the authors to reformulate the text and clearly indicate that the S protein acts only as a priming signal.

We thank reviewer 2 for this important remark. The fact that the S-protein is primarily a priming signal was highlighted in Fig. 1C which illustrates that IL-1b secretion requires both a priming signal and a second signal for execution. Nevertheless, we made sure that the wording was changed accordingly throughout the manuscript. This topic is also clearly presented in the new Fig. 7.

2. Interrogating naive PBMCs, it will be relevant to test whether S protein also result in Aim2 activation following addition of DNA. In other words to test whether the phenotype observed only affects only NLRP3 a or it is a general inflammasome issue. This simple experiment will reinforce the conclusion of S protein as signal 1.

After having discussed this remark extensively with all authors, it is not fully clear to us how to address this question experimentally. In contrast to the NLRP3 inflammasome, which is the major target structure of our work, AIM2 driven inflammasome formation and IL-1b release does not require two signals for full execution.

According to the literature, AIM2 is a sensor of bacterial, viral and endogenous double stranded DNA. Exposure of cells to dsDNA alone leads to AIM2 driven inflammasome activation. We don't see how sequential challenge with the S-protein AND DNA would lead to differential activation of AIM2 or NLRP3. Foreign dsDNA should always activate AIM2 independently from S-protein exposure. We would also like to refer to our extensive RNA-Seq data. The S-protein leads to strong upregulation of NLRP3 but not of AIM2 indicating that the NLRP3 inflammasome is the primary target of S. Indeed, AIM2 was upregulated in unstimulated macrophages from convalescent patients compared to naïve controls. This may show that AIM2 plays a role in SARS-CoV-2 pathogenesis which requires further investigation. However, the S-protein does not seem to be involved.

Nevertheless, we can still create an experiment that addresses the question of reviewer 2 but would require some advice how to set this up.

For clarity and better focus of our manuscript, the sentence of AIM2 being up in SC-conv cells was deleted. The data are still present in the respective Excel files showing all RNA-seq findings.

3. It is remarkable that despite activating NF-kB signaling, transcription of different cytokines and also NLRP3 and AIM2, authors cannot find NLRP3 activation in naive cells upon challenge with the S protein. This is perhaps one of the most surprising findings of the manuscript. Authors need to confirm by immunoblotting the presence of pre IL1 in the lysates of challenged cells (time course experiment). Authors need to show that there are no differences on the interaction of the S protein between different cells. Another experiment to be included is the secretion of TNF following stimulation (in naive and COVID19 cells if the latter is possible).

We thank reviewer 2 for these remarks. We performed the requested immunoblot of pro-IL1b which correlates very well with the qRT-PCR data on IL-1b (Fig. 2F). It becomes evident that the S-protein induces pro-IL-1b expression in both types of macrophages. However, macrophages from naïve individuals fail to activate the NLRP3 inflammasome for caspase 1 activation. IL-1b and GSDMD is not cleaved.

With regard to the remark on differences in different cells and the interaction of the S-protein we would need more advice by the reviewer to address this issue. What is meant exactly: different macrophages in the same well of a 96 well plate? Different types of macrophages? Cell lines? Different kinds of

leukocytes?

TNF secretion occurs independent from the inflammasome and usually does not require a second signal such as nigericin. Many PAMPs such as LPS lead to TNF secretion. We added data on TNF-secretion showing that the S-protein induces secretion of TNF in both healthy control and COVID-19 patient derived cells. Thus, it is only the inflammasome which is differentially regulated. New data can be found in Fig. 1F.

4. The patients' characteristics need to be presented in more detail: treatments received, number of days at the hospital and whether they have received any antibiotic treatment.

This information has been added to the respective table describing hospitalized individuals. It is important to note that a large number of our samples were derived from convalescent or outpatient COVID-19 cases which did not receive any treatment at all.

5. Ferritin is known to activate inflammasome. This might be a confounding factor in the experiments, and may have a long-term impact on macrophage reprogramming. This should be discussed as a limitation.

We agree that ferritin seems to play a role in SARS-CoV-2 pathogenesis as an acute phase protein. At least it can be used as a marker of disease severity. In the revised version of the manuscript we added more information on ferritin levels of hospitalized patients. However, we would like to point out once more that the effects observed in primary macrophages (NLRP3 inflammasome activation, IL-1 β release) were also observed in cells from young outpatients with hardly any symptoms and convalescent patients whose ferritin levels had been normalized probably for many weeks. Given the short half-life of monocytes (2-3 days) we don't see how ferritin (which was at normal plasma levels) should affect inflammasome activation in these cells. Ferritin may have led to reprogramming of monocyte precursor cells in the bone marrow leading to trained innate immunity at earlier time-points. However, we don't see how this would be a limitation of our study. In addition, we now add data on epigenetic reprogramming of monocytes which provide an explanation for the observed effects (cut and run data below)

6. It has been recently published that SARS-CoV-2 induces inflammasome activation in PBMCs (NLRP3 as well as potentially other unknown one(s) *J Exp Med* (2021) 218 (3): e20201707) from naive individuals. This finding using whole virus contradicts the results of this manuscript. This should be discussed as a significant limitation and it should be emphasized the relevance of using the whole pathogen versus purified products.

We thank reviewer 2 for this important remark. We are well aware of this relevant publication which had been cited twice in our initial manuscript (Rodrigues et al., - cited in the introduction and the discussion). This manuscript shows that whole virus exposed to primary macrophages leads to NLRP3 inflammasome formation and IL-1 β release. We now set out to further dissect these important findings by trying to identify the responsible structural components of SARS-CoV-2 which in our view is an important approach in molecular microbiology and infectious diseases research.

When carefully reading the manuscript by Rodrigues et al., we do not see major discrepancies to our work. In fact, whole viral particles did lead to caspase 1 activation and IL-1 β release in naïve macrophages only after addition of a potent TLR2 activating agent (Pam3Cys in this case)(Fig. 1A and B of Rodrigues et al.). Thus, similar to our observations, naïve macrophages required some sort of pre-activation to show the expected effect. In this setting however, whole virus rather functions as a second signal following priming with a TLR2 activating signal. In our view, this does not rule out that SARS-CoV-2 PAMPs may also function as a priming signal.

In summary, we rather believe that both Rodrigues et al. and our manuscript strongly support the

assumption that the NLRP3 inflammasome plays an important role in COVID-19. The data displayed in Rodrigues et al. have now been discussed in our revised manuscript.

7. Authors need to discuss the biological relevance of the system employed: (i) is there any case in vivo in which cells will be challenged with the S protein (extracellularly) but not in the context of the viral particle? (ii) are the levels tested physiological relevant? this information is mostly likely unknown and, therefore, this should be considered as a limitation.

We thank reviewer 2 for this important remark. Biological relevance is always an important issue in ex vivo studies. In our initially submitted manuscript we tried to highlight the fact that the S-protein alone, without the context of a viral infection, is of utmost importance as a vaccine antigen. Thus, we strongly believe that our extensive immunological data on primary macrophages exposed to this protein are highly relevant. Within the context of vaccination, macrophages will indeed be challenged with the S-protein coming from the extracellular micro-environment (as a protein vaccine or after expression from tissue cells). Many of the vaccine constructs currently exploited against SARS-CoV-2 are still experimental. Some vaccines lack potent adjuvants. Thus, we believe that it is important to show that the S-protein in itself induces inflammation. It is very likely that future vaccines will also be based on affinity purified S-protein. Here, human tissue will be challenged with microgram amounts of this protein again highlighting the importance and relevance of our data. The link to the S-protein and vaccination has been described several times in the manuscript.

8. This reviewer does not consider that the tuberculosis data add too much to this work. Is this too imply that the cells need to be activated to sense the S protein and therefore respond?

We would like to thank reviewer 2 for this important remark and regret that we failed to incorporate the tuberculosis data in an understandable way. In comment 3 reviewer 2 states: "It is remarkable that despite activating NF- κ B signaling, transcription of different cytokines and also NLRP3 and AIM2, authors cannot find NLRP3 activation in naive cells upon challenge with the S protein."

We agree that this finding is remarkable and novel. We tried to use the tuberculosis macrophage data to show that this observation is not exclusively dependent on prior SARS-CoV-2 exposure only and that we are observing an unspecific effect. This is indeed highly expected since we are dealing purely with macrophage innate immune functions which need to be relatively unspecific and should not be too discriminative with regard to different pathogens. The tuberculosis data were meant to show that macrophages need to be non-specifically activated by a pathogen that also activates the immune system via TLR2. We believe that this finding is important also for the trained innate immunity aspect of our manuscript which is being exploited as a vaccination approach using BCG. To support this assumption, we provide novel data of S-protein/nigericin treated macrophages from TB patients which received antibiotic treatment for 6 months. While the IL-1 β signal clearly declines, there is a subset of patients whose macrophages still respond (Fig. 6G). This memory response, which seems to be functional for several months is exactly what is anticipated with vaccines that are supposed to induce trained innate immunity (e.g. BCG).

Finally, our novel CUT&RUN (similar to CHIP-seq) datasets provide an additional highly exciting link to trained innate immunity in macrophages from COVID-19 patients and tuberculosis patients. Please see below.

9. The miRNA is reported as a catalogue of genes but it is not mechanistically integrated in the work.

We agree that the issue of trained innate immunity observed in our study requires additional evidence.

We decided to address comment 9 together with comment 10 below.

10. To sustain the statement of long-live reprogramming significant additional evidence needs to be included, including histone analysis and genome-wide *CHIP* data. This reviewer concurs with the impression of the authors; however, in the absence of hard-core data authors need to tone down significantly their statement (including changing the title) and keep this as hypothesis to be validated with further research.

We would like to thank reviewer 2 for this important remark. The concept of trained innate immunity or innate immune memory is relatively new and to date not fully understood. As many aspects in molecular biology and immunity it is most likely a multi-factorial mechanism that leads to profound re-programming of short-lived monocytes. It is believed that both transcriptional and epigenetic reprogramming is involved. There is published evidence that supports the convergence of multiple regulatory layers, including changes in chromatin organization and the persistence of microRNAs (see for example Netea et al, Science 2016). For this reason, we decided not to focus a single miRNA since we found a total of 30 miRNAs which were differentially expressed in SC-conv cells compared to cells from naïve individuals (refers to comment 9). Instead we decided to also look into epigenetic modifications of macrophages and performed an extensive CUT&RUN experiment on SC-conv and SC-naïve peripheral monocytes (CUT&RUN is a whole genome mapping technology similar to ChIP-seq, the method has several advantages over ChIP and works more reliable with low cell numbers). These data clearly demonstrate profound epigenetic changes of histones associated with activation of genes involved in inflammation, inflammation associated activation of myeloid cells and positive regulation of the immune response (Fig. 6). These changes were only observed in SC-conv patient derived cells. Bioinformatic analyses of significantly enriched loci in SC-conv monocytes revealed that these cells are primed towards activation of an immune response associated with infectious diseases (e.g. tuberculosis; see KEGG analysis Fig. 6D and E). This provides an excellent link with our IL-1b secretion data of tuberculosis patient derived cells (see comment 8). On the gene level we were intrigued to see that many of the peaks identified in this genome-wide histone modification analysis were also identified as upregulated genes in the RNA-seq data performed on S-protein stimulated macrophages or non-stimulated macrophages from convalescent individuals (e.g. IL-1b, IL-1a, MYD88, JAK1, CD14, TLR2, S100A8/9/12).

In sum, we now provide strong evidence for inflammation and immune memory associated epigenetic histone modifications in COVID-19 convalescent patient derived monocytes which were absent in cells derived from naïve individuals.

Referee #3 (Remarks for Author):

The manuscript by Theobald et al. studies the response of Covid-naïve versus Covid-exposed monocyte-derived macrophages to the Spike protein of SARS CoV-2. The find distinct gene expression signatures between Covid-naïve and Covid-exposed populations and argue that this shows an innate immune memory that manifests in exacerbated inflammatory responses - mostly centered on IL-1b. The studies are, of course, timely and the available data is important to mine. Unfortunately, the manuscript doesn't solidify its main findings and the result is a series of poorly connected experiments with alternative explanations never ruled out. Enthusiasm for the manuscript in its present form is therefore muted. Major and Minor critiques are as follows:

Major:

1. The comparison of the Spike protein priming the inflammasome is made to LPS, a TLR4 activator. Despite this, the authors show increased expression of TLR2 and then later show that a TLR2 inhibitor blunts the effect of the Spike protein on inflammasome priming. If TLR2 is sensing the Spike protein, shouldn't a TLR2 agonist be the true comparison here, not LPS?

We thank reviewer 3 for taking the time to carefully evaluate our manuscript. We agree that TLR2 agonists should also be tested. However, first we would like to give a general comment. When we started out with this hypothesis driven study it was not clear whether and how the S-protein would induce inflammation in primary human macrophages. In the literature one can find both viral glycoproteins signaling via TLR2 AND TLR4. Thus we decided to use the best described positive control for NLRP3 inflammasome activation which is LPS. It was only after having performed extensive RNA-Seq data with S-protein and LPS exposed macrophages that we came to the assumption that TLR2 may play a role in S-protein driven activation of the inflammasome. Also from the innate immune memory aspect of our manuscript, LPS is an intriguing control since our body is constantly exposed to LPS via gut microbiota which may explain that differential activation of the inflammasome in naïve and SARS-CoV-2 exposed cells only occurs in S-protein primed cells and not in LPS primed cells.

However, we fully agree that TLR2 binding PAMPs represent an interesting control for our observations and added data generated with zymosan, a well described TLR2 agonist. Pilot experiments were performed with both Pam3CSK and Zymosan, the latter clearly gave better signals in our assays. Zymosan was also functional in both macrophages from naïve donors and in macrophages from patients. This stands in contrast to what we observed with S-protein stimulation. Nevertheless, zymosan treated patient macrophages gave higher signals than naïve ones. Overall, the signal is comparable to what we see in LPS treated cells. Thus, in S-protein driven signaling, TLR2 seems to play a role. However, differences in expression levels of TLR2 do not fully explain the differential inflammasome activation we see in patient derived cells compared to those from healthy controls. In the revised manuscript we provide additional mechanistic data on altered REDOX signaling that can induce S-protein driven inflammasome activation, in addition, epigenetic data confirm histone modifications associated with pro-inflammatory cytokine signaling (please see detailed explanation below). New data on zymosan are presented in Fig.4D and E.

2. Along these same lines, the role of TLR2 is hinted at, but never established. Do TLR2-null cells respond to the S protein? Much of the manuscript suffers from this problem - an interesting result is hinted at, but never solidified.

We agree that the role of TLR2 or S-protein receptor binding in general requires further elucidation. However, the suggested experiment is difficult or maybe impossible to perform in primary human macrophages from diseased individuals. CD-14 positive monocytes do not replicate and are short lived. Our protocol requires rapid treatment of monocytes with M-CSF which results in differentiation to macrophages. Shortly afterwards cells are stimulated with PAMPs. To our knowledge, there are no well-established protocols described that achieve gene knock-outs (“null cells”) in such a setting. The gene knock-down technique using siRNA in primary, non-replicating cells is hampered due to immunological off-target effects which are due to various features of the siRNA structure, sequence, and delivery mode (e.g. liposomes), making this method unreliable in assays detecting cytokine secretion. These undesired effects can be overcome by using replicating immortalized macrophage cell lines which allow for a “wash out” phase via multiple cell replication and splitting steps. All experiments performed in this manuscript have been performed with primary human macrophages which is also key to many of our findings (e.g. difference in SC-conv and SC-naïve individuals). In our view, changing to a macrophage cell line (e.g. RAW or THP1) for this experiment would be a sharp methodological break. In addition, our

preliminary experiments performed with macrophage cell lines indicate that their phenotype following S-protein/nigericin stimulation is similar to primary macrophages isolated from SARS-CoV-2 naïve individuals. Thus use of these cell types for downstream experiments is limited.

To still address the issue of TLR2 dependent S-protein priming of diseased primary human macrophages, we decided to stay in our well established experimental setup and performed a blocking experiment with TLR2 neutralizing antibodies. These data demonstrate a significant and dose dependent decline of IL-1b secretion upon treatment with the TLR2 blocking antibody. LPS/nigericin treated cells were used as controls. Here, TLR2 blocking had no effect on IL-1b secretion. Thus we provide several lines of evidence that TLR2 plays a role in S-protein signaling. New data can be found in Fig. 4E

3. Inflammasome activation is never actually shown - Only IL-1b release. Does ASC oligomerize? Is Gasdermin D cleaved? Is Caspase-1 activated?

We agree that inflammasome formation is more than secretion of IL-1b. We had shown secretion of cleaved IL-1b which indirectly shows activation of caspase 1. In addition, we used a chemical genetics approach with well-defined inhibitors of caspase 1 and NLRP3.

In our revised version we now show ASC speck formation which is a hallmark of NLRP3 inflammasome formation. Exploiting fluorescent microscopy, we show that ASC assembles into a large protein complex after stimulation with the S-protein/nigericin. This occurs to a higher number in patient derived cells (Fig. 2C,D). We were also able to show cleavage of GSDMD upon incubation with the S-protein (Fig. 2H). Unfortunately, we had technical difficulties detecting caspase 1 in our primary human macrophage model despite using different commercially available antibodies. However, with the many additional data presented in this revised manuscript (e.g. two cleaved substrates of caspase 1) we hope that reviewer 3 can appreciate that NLRP3 inflammasome formation does occur.

4. There is bimodal distribution in the response to the Spike protein. This can be seen in Figure 1D (last bar - bar 10) and in Figure 2A - DMSO treated. From the data shown in Figure 2G, those two patients studied must have been the highest responders in 1D and 2G. Given this, it isn't clear that patients generally respond as in 2G. This again is an interesting finding that isn't solidified.

We agree that patients selected for 2G had relatively high IL-1b levels at baseline. This figure should illustrate that the response declines over time. With regard to the relatively large range of IL-1b values we observe at single time-point experiments (Fig. 1D) we would like to point to the fact that we are working with patient derived cells which are not matched with regard to age, sex and exact time-point of infection as this would be the case in an animal experiment. Thus, we would be hesitant to describe the response shown in Fig. 1D as bimodal. This is best illustrated by the LPS data. LPS is a very well described PAMP activating a multitude of pro-inflammatory pathways. This should happen in both, healthy and diseased individuals. We can confirm this for IL-1b secretion in our assay and there clearly is a statistically highly significant difference in LPS treated cells compared to untreated cells. However, also in LPS treated cells, there is a large range of IL-1b values with a relatively large SD. We hope that reviewer 2 will agree that this argues against a bimodal response for both LPS and S but rather for the expected inter-individual differences of the immune response in patient derived biomaterial.

Nevertheless, to better illustrate that we do have increased IL-1b secretion in all diseased versus healthy S-protein treated cells and in S-protein treated versus untreated cells, we created a figure where data-points are connected with lines. Supplementary fig. 1E (log scale!).

5. The gene expression data is parsed in a number of ways, but the in vivo effect of the changes in gene expression isn't shown. For instance, in Figure 3C, it shows that MAPK signaling is severely altered, but this is never studied. These sorts of experiments are hypothesis-generating, but become important when those hypotheses are tested. Does MAPK signaling affect the Covid inflammatory response? Almost every pathway shown could be studied. Instead, the data is presented but not in the context of a follow up experiment that shows its role in Covid inflammation. The same critique could be made of Figure 4.

We agree that omics data such as RNA-Seq provide a large amount of information that requires proper interpretation. We generated RNA-Seq datasets for several reasons.

1. *To confirm that the S-protein leads to strong upregulation of inflammasome associated genes*
2. *To delineate the S-protein triggered gene expression pattern from other inflammasome activating PAMPs*
3. *To understand why macrophages from COVID-19 patients/convalescent indiv. react so fundamentally different to S-protein stimulation compared to those from naïve individuals*

With regard to topic 1, we believe that the RNA-Seq data nicely confirm that the S-protein strongly targets NF-κB driven upregulation of inflammasome associated genes. Among the top genes with multiple fold upregulation is e.g. IL-1b, IL-18, NLRP3 and so on. It is true that, like many other PAMPs, the S-protein also regulates a multitude of other genes (though to a much lesser extent than LPS). An alternative approach would be to focus and not to show these data at all. However, COVID-19 is an important disease and the S-protein a key protein in the fight against COVID-19, thus we felt it would be very helpful to provide the full dataset to the many researchers involved in the fight against this infectious disease. Since reviewer 3 asks for MAPK in general we tested a potent p38 MAPK inhibitor (doramapimod, 10 uM) in our inflammasome assay. P38 MAPK is a central hub regulating many immune functions in the cell. The substance failed to inhibit IL-1b secretion stimulated by the S-protein. It is likely that other cytokines stimulated by the S-protein may be regulated by MAPKs (e.g. TNF or IL6). Since the inflammasome and IL-1b secretion is the main topic of our paper, we would prefer not to add these data to the manuscript.

With regard to topic 2 and 3 we simply used quantitative gene expression data to show that naïve cells show much less S-protein driven differential gene expression than cells from COVID-19 patients. This is not the case for LPS exposed cells. This hints towards a global pattern of reprogramming in COVID-19 derived cells which renders these cells more responsive to a certain PAMP than to another PAMP. We believe that this is an intriguing and novel finding which should be presented.

Since reviewer 3 asks for a follow up experiment for our RNA-Seq findings we had a closer look on the gene expression profile of unstimulated macrophages. As already pointed out in our manuscript, the gene expression pattern of COVID-19 convalescent patients strongly hints towards dysregulation in genes involved in oxidative stress. Thus we speculated that oxidative stress induced by small molecules may render macrophages from COVID-19 naïve individuals responsive towards S-protein priming. In this experimental setup we pre-treated macrophages with low amounts of H2O2 or FCCP to induce oxidative stress. Intriguingly, this treatment led to a measurable increase in IL-1b upon S-protein/nigericin stimulation similar to cells isolated from COVID-19 patients (Fig. 5E).

We assume that COVID-19 (and other infectious agents) lead to modifications in the intracellular redox responsiveness when stressed with foreign antigens allowing for differential reactivity towards certain inflammasome activating PAMPs.

These new datasets are based on careful interpretation of global gene expression data which we believe are also very useful for other researchers performing follow-up experiments in the field of innate immunity.

6. The argument is that Covid induces a sort of innate immune memory. Mechanistically, for reasons that are unclear, this is attributed to miR-RNA expression (Figure 5). This is never experimentally tested however. It isn't unexpected that miRs change upon Covid exposure - that doesn't prove miRs are causing this though. This needs to be shown. As an aside, I would think an epigenetic phenomenon would be more likely.

This is an important comment and we also agree that the concept of trained innate immunity is most likely driven by several layers of gene regulation. It is believed that both transcriptional and epigenetic reprogramming is involved. There is published evidence that supports the convergence of multiple regulatory layers, including changes in chromatin organization and the persistence of microRNAs (see for example Netea et al, Science 2016). For this reason, we decided not to focus a single miRNA since we found a total of 30 miRNAs which were differentially expressed in SC-conv cells compared to cells from naïve individuals. Our understanding is that in innate immune memory, miRNAs rather provide a signature that is associated with this phenomenon and that additional transcriptomic and epigenetic

data are required to confirm memory in these short lived cells. As suggested by reviewer 3, we decided to also look into epigenetic modifications of macrophages and performed an extensive CUT&RUN experiment on peripheral blood monocytes from SC-conv and SC-naïve individuals (CUT&RUN is a whole genome mapping technology similar to ChIP-seq, the method has several advantages over ChIP and works more reliable with low cell numbers).

These novel data clearly demonstrate profound epigenetic changes of histones associated with activation of genes involved in inflammation, inflammation associated activation of myeloid cells and positive regulation of the immune response (Fig. 6). These changes were only observed in SC-conv patient derived cells. Bioinformatic analyses of significantly enriched loci in SC-conv monocytes revealed that these cells are primed towards activation of an immune response associated with infectious diseases (e.g. tuberculosis; see KEGG analysis Fig. 6D and E). This provides an excellent link with our IL-1b secretion data of tuberculosis patient derived cells (see comment 7). On the gene level we were intrigued to see that many of the peaks enriched in this genome-wide histone modification analysis were also identified as upregulated genes in the RNA-seq data performed on S-protein stimulated macrophages or non-stimulated macrophages from convalescent individuals (e.g. IL-1b, IL-1a, MYD88, JAK1, CD14, TLR2, S100A8/9/12).

In sum, we now provide strong evidence for inflammation and immune memory associated epigenetic histone modifications in COVID-19 convalescent patient derived monocytes which were absent in cells derived from naïve individuals.

7. The Tb data is thrown into the manuscript without much context. Patients with acute TB respond to the S protein, but in context of the manuscript is the question of whether recovered Tb patients respond to the S protein. What about other infections like influenza or patients with chronic inflammatory disease? Again, this interesting finding is never verified or studied in depth.

We regret that we failed to incorporate the tuberculosis data in an understandable way.

Initially our data suggested that the S-protein functions as a primer antigen selectively initiating IL-1b release dependent on previous SARS-CoV-2 exposure.

We tried to use the tuberculosis macrophage data to show that our observations are not exclusively dependent on prior SARS-CoV-2 exposure and that we are observing an unspecific effect. This is indeed highly expected since we are purely dealing with macrophage innate immune functions which need to be relatively unspecific and should not be too discriminative with regard to different pathogens. The tuberculosis data were meant to show that macrophages need to be non-specifically activated by a pathogen that also activates the innate immune system via toll like receptors.

We are grateful for the suggestion of using recovered patients. We exploited our longitudinal TB biobank and retested patients that had received 6 months of anti-TB treatment. All patients are culture confirmed Mtb positive cases receiving a suitable antibiotic treatment according to the resistance profile of the respective strain. In a proportion of the treated patients we observed a clear decline of IL-1b levels upon S-protein/nigericin stimulation (Fig. 6G). Intriguingly though, there was a subset of patients that remained reactive at relatively high levels. This clearly shows that immune memory of macrophages can be long lasting, existing for several months depending on the nature of the stimulatory agent. A finding that may support the usage of BCG vaccines as boosters for the innate immune system.

We agree with reviewer 3 that the highly interesting discovery of extremely tightly controlled inflammasome activation which seems to be selectively primed when using different PAMPs requires follow up studies with other pathogens and inflammatory diseases. In this manuscript we focused on Mtb and SARS-CoV-2. The data we present are based on labor-intensive biobanking of clinical samples over the past year(s) which requires extensive resources in both the clinic and the laboratory. In the future, we

will extent our efforts to other disease backgrounds to address these highly interesting research questions in the future. Interestingly, this flu season, we hardly see any influenza patients in our clinic which is in itself an intriguing observation and most likely due to the strict lockdown and usage of protective gear in the entire population.

Minor

1. Pro-IL1b needs to be shown in Western blots (esp 2D). This is especially true as Figure 1C implies that Covid-exposed macrophages have higher basal levels of pro-IL1b and might be primed at baseline - not only at S protein exposure.

We performed the requested immunoblot of pro-IL1b which correlates very well with the qRT-PCR data of IL-1b (Fig. 2F). It becomes evident that the S-protein induces pro-IL-1b expression in both types of macrophages. However, macrophages from naïve individuals fail to activate the NLRP3 inflammasome for caspase activation. IL-1b and GSDMD are not cleaved in these cells.

2. It isn't clear how Western blots are quantified. They are studying supernatant but quantifying it to an area on the gel with no signal? It needs to be quantified to something.

For immunoblots performed with cell lysates we correlated band intensity to the beta-actin signal. For supernatants of cells undergoing pyroptosis upon NLRP3/Casp1 activation, a similar approach is not feasible (see GAPDH immunoblot performed for rev. 1). We quantified the signal obtained from cell supernatants to the number of input cells added to the respective wells in the experiment. This is now described in the methods section.

3. The figure presentation in Figure 2 is confusing. The box explaining the colors doesn't apply to every panel and its very distracting. Each panel should be outlining what's being studied.

We thank reviewer 3 for this important point. The labels now explain every single panel of the fig.

4. Why is the IL-1b released in Figure 3H so much lower in Figures 1D or 2F?

We assume that reviewer 3 refers to fig. 3G? Data using the TLR2 inhibitor in S-primed patients? We added more data points from additional patients with higher values.

Please note that most data are based on samples from patients, there is an expected degree of variation in cytokine responses. This is equally observed in the LPS control. Note that there the overall numbers of individuals tested in our study strongly increased in our revised manuscript. Key findings are now supported with more datapoints.

5. The authors note the inflammasome-related expression of Pycard and Caspase-8 mRNA is significantly down in Figure 3D. Shouldn't this blunt the inflammasome (especially pycard (ASC).

We thank reviewer 3 for addressing this observation. It is likely that at an earlier or later time-point, ASC may be expressed to higher levels. Very little is known about timing of gene regulation of inflammasome components. ASC is activated by oligomerization which may not require gene expression to higher levels. In our revised manuscript we present ASC speck formation upon S-priming clearly showing that the protein is present and activated.

In summary, while the manuscript is timely and there are some interesting findings, none of these

findings is studied in sufficient depth to be convincing. This fact, coupled with numerous experimental issues, significantly diminishes enthusiasm for the manuscript.

We strongly believe that with the many additional data we now provide in this revised manuscript we provide sufficient experimental evidence to support the key messages presented in this paper. Relevance of Omics data was now confirmed with follow up experiments. State of the art experiments strongly support NLRP3 inflammasome activation in patient derived macrophages.

We would like to thank reviewer 3 once more for taking the time reading our paper and for providing relevant points of critique and suggestions for additional experiments which helped to improve our manuscript.

8th Mar 2021

Dear Dr. Rybniker,

Thank you for the re-submission of your manuscript to EMBO Molecular Medicine. We have now received feedback from the three reviewers who agreed to evaluate your manuscript. As you will see from the reports below, all three referees are supporting the publication of the manuscript, however, referee #1 raises some critique that requires additional experimentation. After our cross-commenting session it became clear that additional experiments should assess NLRP3 phosphorylation and ubiquitination status and the status of NEK7 in virus-experienced versus naive macrophages.

Addressing the reviewers' concerns in full will be necessary for further considering the manuscript in our journal, and acceptance of the manuscript will entail a second round of review. EMBO Molecular Medicine encourages a single round of revision only and therefore, acceptance or rejection of the manuscript will depend on the completeness of your responses included in the next, final version of the manuscript. For this reason, and to save you from any frustrations in the end, I would strongly advise against returning an incomplete revision.

We realize that the current situation is exceptional on the account of the COVID-19/SARS-CoV-2 pandemic. Therefore, please let us know if you need more than three months to revise the manuscript.

I look forward to receiving your revised manuscript.

Yours sincerely,

Zeljko Durdevic

***** Reviewer's comments *****

Referee #1 (Comments on Novelty/Model System for Author):

The system probed by the authors is relevant. The reagents and tools are well established in the field.

Referee #1 (Remarks for Author):

I commend the authors for their efforts to meet the issues raised by the reviewers. This reviewer still strongly believes that the connection with trained immunity is not mechanistically done. Curiously, it is not necessary to make the case for the research presented. Likewise for the tuberculosis data. In addition, there are few other points that require clarification:

1. Fig 1D and 1E. Both figures are reporting essentially the same, and I would suggest that Fig 1E may be better reported as supplementary material.
2. Fig 2F and sup 2E. Data in the supplementary figure does not include any statistics, making difficult to argue that indeed there are differences in the levels of pro IL1b. Legend 2F refers to "three individual experiments". Does this mean three different patients? or three repetitions of the same patient? The latter is not enough to support the authors' claim. If it is the former, then this reviewer urges the authors to present also a quantification analysis (using ImageJ plus) of the three different blots.
3. The TLR2 experiments have solidified the work greatly. However, still it may be argue that there are differences in the levels of the binding of the protein to the receptor. Authors have not quantified the TLR2 surface levels (by flow cytometry for example). The zymosan experiments mitigate this limitation. Nevertheless, the golden standard experiment, the quantification of the receptor, is not yet included. This should be acknowledged as a limitation in the discussion.
4. It is intriguing that despite the priming signal is there, ie S protein did induce proIL1B, NLRP3 as well as other inflammatory mediators, there is no ASC spec formation and secretion of iL1b. Providing mechanistic insights to these observations would have dramatically increased the enthusiasm for this work. The results included in Fig5E are a good starting point but still fall short of in-depth molecular evidence. It would have important to investigate the interaction between NLRP3 and ASC, the ubiquitination of NLRP3, the status of NEK7 just to mention few of the well known elements affecting NLRP3 activation. And compare all these with Covid19 cells since the system is working when these cells are challenged.
5. Authors need to clarify how they are able to detect the intracellular pro-IL1b with the same E:LISA used to detect the mature iL1b in the supernatants. How can they differentiate the still non secreted IL1b versus the pro-IL1b?

Referee #2 (Comments on Novelty/Model System for Author):

These studies appropriately used primary monocyte-derived human macrophages (generated by 5-days ex vivo differentiation in the presence of m-CSF) that were isolated from multiple cohorts of COVID-19 patients (stratified by disease severity and convalescent state), control (SARS-CoV2-naive) subjects, and Mycobacteria tuberculosis patients. The primary macrophages were rigorously assayed for functional inflammasome signaling, transcriptional expression profiles, and epigenetic profiling of histone marks associated with chromatin modifications in myeloid leukocytes displayed trained immunity.

Referee #2 (Remarks for Author):

General Comments: This extensively revised MS describes the ability of recombinant SARS-CoV-2 spike protein (S-protein) to act as an ex vivo "signal 1" priming stimulus for NLRP3 inflammasome assembly in human macrophages differentiated from the blood monocytes of COVID-19 patients but not virus naive "control" donors. As noted in my initial review, the data support a model of trained innate immunity whereby myeloid progenitors are epigenetically reprogrammed during in vivo S-protein exposure to stably upregulate sensors of S-protein. The original MS presented

functional analyses and transcriptomic screens which support elevated expression of TLR2 as a potential S-protein receptor that can drive NFkB-dependent induction of NLRP3 and IL1-b as "primed" components of the NLRP3 inflammasome signaling axis in monocyte/macrophages from COVID-19 patients. As a consequence, exposure of these primed macrophages to canonical "signal 2" stimuli which disrupt ion homeostasis (e.g., nigericin or ATP-activated P2X7 receptor channels) facilitates rapid assembly of NLRP3/caspase-1 inflammasome with consequent proteolytic maturation and release of IL-1b, a known component of the inflammatory "cytokine storm" in COVID-19 patients. Other data indicate that: 1) "trained" monocyte/macrophages from convalescent COVID-19 subjects are also defined by altered expression of various miRNA's associated with enhanced proinflammatory tone; and 2) monocytes/macrophages isolated from MTB patients are defined by a similar trained innate response to an ex vivo S-protein priming of the NLRP3 inflammasome signaling axis.

In response to the comments and suggestions from the 3 reviewers of the initial submission, the authors now provide much more mechanistic support for the major conclusions and a clearer exposition of the overall model. The new CUT&RUN analyses of trimethylated K4 in histone 3, and acetylated K27 of that histone, in Fig 6 are particularly strong and consistent with the recently described epigenetic modifications that define trained immunity of myeloid cells. The proposed role(s) of altered intracellular redox milieu (secondary to changes in metallothionein and S100A8/A9 expression) in coupling TLR2 activation to the likely post-transcriptional modification(s) of NLRP3 inflammasome components are now more clearly presented and conceptually plausible.

Thus, the revised MS provides timely, relevant and mechanistically compelling findings. The findings and model set the foundation for future analyses of whether similar training of inflammasome signaling is observed in the monocyte/macrophages from healthy subjects who have received the various spike protein-based vaccines.

Referee #3 (Comments on Novelty/Model System for Author):

I think the technical quality, the novelty and the adequacy of the system are very strong. Medical impact would be high if they were able to show that patients could be stratified to receive anti-IL-1 to decrease severity. This is far outside the scope of what is reasonable to ask, however.

Referee #3 (Remarks for Author):

I think the authors made a good faith effort to address my critiques and the manuscript is much stronger as a result. I particularly like the results of the epigenetic experiments at the tail end of the manuscript. I think this adds a level of global mechanistic depth that was missing initially. Strong work!

Rebuttal for EMM-2021-14150 “Long-lived macrophage reprogramming drives spike protein-mediated inflammasome activation in COVID-19”

Referee #1 (Comments on Novelty/Model System for Author):

The system probed by the authors is relevant. The reagents and tools are well established in the field.

We would like to thank reviewer 1 for thoroughly evaluating our revised manuscript and for providing suggestions helping to improve our data. A detailed point by point response is provided below.

Referee #1 (Remarks for Author):

I commend the authors for their efforts to meet the issues raised by the reviewers. This reviewer still strongly believes that the connection with trained immunity is not mechanistically done. Curiously, it is not necessary to make the case for the research presented. Likewise for the tuberculosis data. In addition, there are few other points that require clarification:

1. Fig 1D and 1E. Both figures are reporting essentially the same, and I would suggest that Fig 1E may be better reported as supplementary material.

We agree with this suggestion and added Fig. 1E to the supplement. Now Figure EV1A

2. Fig 2F and sup 2E. Data in the supplementary figure does not include any statistics, making difficult to argue that indeed there are differences in the levels of pro IL1b. Legend 2F refers to "three individual experiments". Does this mean three different patients? or three repetitions of the same patient? The latter is not enough to support the authors' claim. If it is the former, then this reviewer urges the authors to present also a quantification analysis (using ImageJ plus) of the three different blots.

We thank reviewer 1 for this comment which will help to present our data in a better way. We performed the blots from 3 individual study participants derived from each experimental group (SC-naive n=3 and COVID-19 n=3). In fact, all data-points in our manuscript are derived from individual patients and not from replicates of a single patient. This is now clarified in the manuscript.

We quantified the blots. The data have been added to the manuscript in figure EV1G.

3. The TLR2 experiments have solidified the work greatly. However, still it may be argue that there are differences in the levels of the binding of the protein to the receptor. Authors have not quantified the TLR2 surface levels (by flow cytometry for example). The zymosan experiments mitigate this limitation. Nevertheless, the golden standard experiment, the quantification of the receptor, is not yet included. This should be acknowledged as a limitation in the discussion.

We thank Reviewer 1 for this comment. We included data on TRL2 levels in Fig. 4A. FACS analysis shows that TLR2 is significantly upregulated on COVID-19 patient derived macrophages which partly explains the selective response to the S-protein. TLR2 is also upregulated to higher levels in S-protein treated COVID-19 patient cells. The respective gene shows histone modifications in our CUT&RUN experiments.

4. It is intriguing that despite the priming signal is there, ie S protein did induce proIL1B, NLRP3 as well as other inflammatory mediators, there is no ASC spec formation and secretion of iL1b. Providing mechanistic insights to these observations would have dramatically increased the enthusiasm for this work. The results

included in Fig5E are a good starting point but still fall short of in-depth molecular evidence. It would have important to investigate the interaction between NLRP3 and ASC, the ubiquitination of NLRP3, the status of NEK7 just to mention few of the well known elements affecting NLRP3 activation. And compare all these with Covid19 cells since the system is working when these cells are challenged.

We agree with reviewer 1 that inflammasome regulation is complex and involves several layers. A multitude of regulatory proteins and pathways seem to influence this complex multiprotein structure. We assessed the suggested regulatory layers with additional experiments:

- 1. We quantified NEK7 in macrophages from COVID-19 patients and clearly found higher levels of this important protein compared to macrophages from naïve controls. Stimulation with either LPS or S-protein for 4h and additional 2h with nigericin did not further increase NEK7 levels which is in line with published data (Shi et al., Nat. Immunology 2016). Elevated levels of NEK7 have also been found in other inflammatory diseases (Chen et al., Cell Death and Disease 2019). With NEK7, we identified an additional inflammasome associated protein that is expressed to higher levels in COVID-19 patients. Elevated baseline NEK7 protein levels may contribute to the differential inflammasome activation and formation of ASC speck (Manuscript Fig. 2C/D) we observe in patients versus controls. Novel data are presented in Figure EV1G,F and below.*

Figure 1

Figure 1: Detection of NEK7 (35 kDa, top row) in total cell lysates from healthy (upper panel) and COVID-19 (lower panel) macrophages. Macrophages were stimulated with LPS or S-protein (4h) followed by 2h of nigericin treatment. Control cells were left unstimulated and β -actin (45 kDa) was used as loading control.

- 2. We performed several experiments with multiple patients and SARS-CoV-2 naïve individuals looking into post-translational modification of NLRP3. We first performed NLRP3 pull-down experiments with human macrophage lysates and determined the ubiquitination status of this protein using immuno-blot. According to published data which are primarily based on mouse models and cell lines, higher levels of ubiquitination have an inhibitory effect on NLRP3 inflammasome formation and we would have expected extensive ubiquitination of NLRP3 in SARS-CoV-2 naïve controls if this would be the primary regulatory mechanism in our test system. However, we were not able to detect the expected ubiquitination pattern in macrophage-derived NLRP3 (Fig. 2A below). In addition, stimulation of macrophages with LPS/nigericin or S-protein/nigericin did not change ubiquitination levels of NLRP3. To confirm that the NLRP3 ubiquitination status can, in principal, be modified in human macrophages, we treated*

macrophages with the ubiquitinase isopaptidase inhibitor G5 (see also By et al., Mol Cell 2013). As expected, G5 treatment led to increased levels of NLRP3 ubiquitination and this inhibitor experiment functions as a positive control for our experiments (Fig. 2B). However, even when treated with G5, there was no differential ubiquitination status detected in S-protein treated patient or naïve control cells. Finally, we determined the phosphorylation status of NLRP3 since phosphorylation of this protein has been linked to ubiquitination. Using a phospho-NLRP3 antibody we were not able to detect differential phosphorylation in human macrophages (Fig. 2C).

Figure 2

Figure 2: A Detection of upiquitinated NLRP3 in cell lysates from healthy (left) and COVID-19 (right) macrophages. Macrophages were stimulated with LPS or S-protein (4h) followed by 2h of nigericin treatment. Control cells were left unstimulated. NLRP3 (bottom) was detected in order to verify immune precipitation of the correct protein. Representative example of three individual experiments. **B** Detection of upiquitinated NLRP3 (top) total cell lysates from primary macrophages. Macrophages were stimulated with S-protein (4h) followed by 2h of nigericin treatment. 15min prior to nigericin the ubiquitinase isopaptidase inhibitor G5 (2μM) was added. Control cells were left unstimulated. NLRP3 (bottom) was detected in order to verify immune precipitation of the correct protein. Representative example of two individual experiments. Ubiquitin Antibody detects ubiquitin, polyubiquitin and ubiquitinated proteins (1:1000 dilution). **C** Detection of p-NLRP3 (110kDa, top row, antibody targets NLRP3 around the phosphorylation site of Ser295, dilution 1:1000) in total cell lysates from healthy (upper panel) and COVID-19 (lower panel) macrophages. Macrophages were stimulated with LPS or S-protein (4h) followed by 2h of nigericin treatment. Control cells were left unstimulated and β-actin (45kDa) was used as loading control. Representative example of three individual experiments.

We also performed an extensive literature search on the control of the inflammasome by the ubiquitin system. There is evidence that in cells exposed to inflammasome activating signaling molecules (LPS and ATP/nigericin), differential NLRP3 ubiquitination is only observed when a deubiquitinase is inhibited or deleted. Py et al. showed that the deubiquitinase BRCC3 needs to be blocked to achieve NLRP3 ubiquitination (Molecular Cell, 2013, Fig. 3C). Palazon-Riquelme et al. show the same for chemical USP7 inhibition (EMBO reports, 2018, Fig. 4B). Using our genome wide RNA-seq data of COVID-19 patient derived macrophages we looked for differential transcriptional levels of deubiquitinases and ligases linked to NLRP3 modification (USP7, USP47, BRCC3, ABRO1, TRIM31, ...). None of these proteins was differentially regulated in COVID-19 patient derived macrophages compared to controls.

To conclude, we did not find evidence for differential post-translational modification of NLRP3 in human macrophages of COVID-19 patients. It is conceivable that posttranslational modification of NLRP3 (e.g. ubiquitination) is required to shut down the inflammasome several weeks after exposure to SARS-CoV-2. This is now discussed in our manuscript. However, we found strong evidence for differential inflammasome regulation on the transcriptional level which seems to involve the entire signaling cascade starting on the receptor level (most likely TLR2). TLR2, NEK7 and pro-IL-1b were upregulated in COVID-19 patients at baseline without any stimulation with PAMPs. TLR2 and IL-1b were further upregulated upon stimulation with the S-protein. And most intriguingly, key components of the cascade leading to processing and release of IL-1b were upregulated to significantly higher levels in S-protein stimulated COVID-19 patient derived macrophages compared to stimulated naïve cells (Caspase 1, GSDMD, NLRP3, TLR2).

Here, we would like to point out again that LPS, a PAMP that stimulates IL-1b release independent from prior in vivo stimulation did not lead to differential expression of these components in COVID-19 patient derived cells compared to cells from naïve controls. This highlights again the importance of differential gene regulation for the observed effects in S-protein stimulated cells.

For most of the tested proteins, differential regulation was confirmed for both RNA-level (qRT-PCR, RNAseq) and protein level (immuno-blot, FACS). We also went further upstream to the DNA level to show that COVID-19 has profound impact on histone modification at exactly the same set of genes which is an excellent explanation for the different gene expression levels we found. Thus we strongly believe that a meaningful link to trained innate immunity has been made since induction of a trained immune phenotype enables cells to react with stronger, more rapid or qualitatively different transcriptional responses when challenged with subsequent triggers. This is exactly what we observe in our test system.

We believe that in this revised manuscript we provide sufficient evidence for differential inflammasome activation which primarily occurs on the transcriptional level. It is likely that there are additional layers of NLRP3 regulation in diseased macrophages. These require further evaluation in future studies.

Phosphorylation and ubiquitination data can be added to the manuscript if required.

5. Authors need to clarify how they are able to detect the intracellular pro-IL1b with the same ELISA used to detect the mature iL1b in the supernatants. How can they differentiate the still non secreted IL1b versus the pro-IL1b?

We thank reviewer 1 for this important remark and for highlighting this labeling error. The ELISA system used in our study detects both full length and cleaved IL-1b (total IL-1b). We assume that in cell lysates we

primarily detect pro-IL-1b with this ELISA. We added this quantitative dataset since it nicely supports our immuno-blot data (Fig. 2F) specifically detecting pro-IL-1b as well as our qRT-PCR data. All datasets support the assumption that there are elevated baseline levels of IL-1b in COVID-19 patient derived cells. We changed the labelling of this supplementary figure.

Referee #2 (Comments on Novelty/Model System for Author):

These studies appropriately used primary monocyte-derived human macrophages (generated by 5-days ex vivo differentiation in the presence of m-CSF) that were isolated from multiple cohorts of COVID-19 patients (stratified by disease severity and convalescent state), control (SARS-CoV2-naive) subjects, and Mycobacteria tuberculosis patients. The primary macrophages were rigorously assayed for functional inflammasome signaling, transcriptional expression profiles, and epigenetic profiling of histone marks associated with chromatin modifications in myeloid leukocytes displayed trained immunity.

Referee #2 (Remarks for Author):

General Comments: This extensively revised MS describes the ability of recombinant SARS-CoV-2 spike protein (S-protein) to act as an ex vivo "signal 1" priming stimulus for NLRP3 inflammasome assembly in human macrophages differentiated from the blood monocytes of COVID-19 patients but not virus naive "control" donors. As noted in my initial review, the data support a model of trained innate immunity whereby myeloid progenitors are epigenetically reprogrammed during in vivo S-protein exposure to stably upregulate sensors of S-protein. The original MS presented functional analyses and transcriptomic screens which support elevated expression of TLR2 as a potential S-protein receptor that can drive NFkB-dependent induction of NLRP3 and IL1-b as "primed" components of the NLRP3 inflammasome signaling axis in monocyte/macrophages from COVID-19 patients. As a consequence, exposure of these primed macrophages to canonical "signal 2" stimuli which disrupt ion homeostasis (e.g., nigericin or ATP-activated P2X7 receptor channels) facilitates rapid assembly of NLRP3/caspase-1 inflammasome with consequent proteolytic maturation and release of IL-1b, a known component of the inflammatory "cytokine storm" in COVID-19 patients. Other data indicate that: 1) "trained" monocyte/macrophages from convalescent COVID-19 subjects are also defined by altered expression of various miRNA's associated with enhanced proinflammatory tone; and 2) monocytes/macrophages isolated from MTB patients are defined by a similar trained innate response to an ex vivo S-protein priming of the NLRP3 inflammasome signaling axis.

In response to the comments and suggestions from the 3 reviewers of the initial submission, the authors now provide much more mechanistic support for the major conclusions and a clearer exposition of the overall model. The new CUT&RUN analyses of trimethylated K4 in histone 3, and acetylated K27 of that histone, in Fig 6 are particularly strong and consistent with the recently described epigenetic modifications that define trained immunity of myeloid cells. The proposed role(s) of altered intracellular redox milieu (secondary to changes in metallothioneine and S100A8/A9 expression) in coupling TLR2 activation to the likely post-transcriptional modification(s) of NLRP3 inflammasome components are now more clearly presented and conceptually plausible.

Thus, the revised MS provides timely, relevant and mechanistically compelling findings. The findings and model set the foundation for future analyses of whether similar training of inflammasome signaling is observed in the monocyte/macrophages from healthy subjects who have received the various spike

protein-based vaccines.

We thank reviewer 2 for evaluating our manuscript a second time and for providing these positive and encouraging comments.

Referee #3 (Comments on Novelty/Model System for Author):

I think the technical quality, the novelty and the adequacy of the system are very strong. Medical impact would be high if they were able to show that patients could be stratified to receive anti-IL-1 to decrease severity. This is far outside the scope of what is reasonable to ask, however.

Referee #3 (Remarks for Author):

I think the authors made a good faith effort to address my critiques and the manuscript is much stronger as a result. I particularly like the results of the epigenetic experiments at the tail end of the manuscript. I think this adds a level of global mechanistic depth that was missing initially. Strong work!

We thank reviewer 3 for these positive comments.

20th May 2021

Dear Dr. Rybniker,

Thank you for the submission of your revised manuscript to EMBO Molecular Medicine. I am pleased to inform you that we will be able to accept your manuscript pending the following final amendments:

- 1) Please include phosphorylation and ubiquitination data as suggested by the referee.
- 2) In the main manuscript file, please do the following:
 - Correct/answer the track changes suggested by our data editors by working from the attached/uploaded document.
 - Add up to 5 keywords.
 - Remove text highlight colour.
 - Add callouts for Fig EV3A and callout Fig EV1M before Fig EV2.
 - In M&M, a statistical paragraph that should reflect all information that you have filled in the Authors Checklist, especially regarding randomization, blinding, replication.
 - Indicate in legends exact $p=$ values, not a range, along with the statistical test used. To keep the figures "clear" some authors found providing an Appendix table Sx with all exact p -values preferable. You are welcome to do this if you want to.
 - Include a statement that informed consent was obtained from all human subjects and that, in addition to the WMA Declaration of Helsinki, the experiments conformed to the principles set out in the Department of Health and Human Services Belmont Report.
 - Please merge "Funding" section with "Acknowledgements".
 - Add author contributions for Jason Chen, Florian Erger and Kanika Vanshylla. If G.T stands for Theodoros Georgomanolis, it should be changed to T.G.
 - In the reference list, where there are more than 10 authors on a paper, 10 will be listed, followed by "et al.". Please check "Author Guidelines" for more information.
<https://www.embopress.org/page/journal/17574684/authorguide#referencesformat>
 - Please be aware that all datasets should be made freely available upon acceptance, without restriction.
Please check "Author Guidelines" for more information.
<https://www.embopress.org/page/journal/17574684/authorguide#availabilityofpublishedmaterial>
-
- 3) Dataset: To the zipp file please add a legend/short description of the datasets as a .doc document.
- 4) The Paper Explained: Please add it to the main manuscript text.
- 5) Source data: We encourage you to include the source data for figure panels that show essential data. Numerical data should be provided as individual .xls or .csv files (including a tab describing the data). For blots or microscopy, uncropped images should be submitted (using a zip archive if multiple images need to be supplied for one panel). Please check "Author Guidelines" for more information. <https://www.embopress.org/page/journal/17574684/authorguide#sourcedata>
- 6) Press release: Please inform us as soon as possible and latest at the time of submission of the revised manuscript if you plan a press release for your article so that our publisher could coordinate publication accordingly.
- 7) Please be aware that we use a unique publishing workflow for COVID-19 papers: a non-typeset PDF of the accepted manuscript is published as "Just Accepted" on our website. With respect to a possible press release, we have the option to not post the "Just Accepted" version if you prefer to

wait with the press release for the typeset version. Please let us know whether you agree to publication of a "Just accepted" version or you prefer to wait for the typeset version.

8) As part of the EMBO Publications transparent editorial process initiative (see our Editorial at <http://embomolmed.embopress.org/content/2/9/329>), EMBO Molecular Medicine will publish online a Review Process File (RPF) to accompany accepted manuscripts. This file will be published in conjunction with your paper and will include the anonymous referee reports, your point-by-point response and all pertinent correspondence relating to the manuscript. Let us know whether you agree with the publication of the RPF and as here, if you want to remove or not any figures from it prior to publication. Please note that the Authors checklist will be published at the end of the RPF.

9) Please provide a point-by-point letter INCLUDING my comments as well as the reviewer's reports and your detailed responses (as Word file).

I look forward to reading a new revised version of your manuscript as soon as possible.

Yours sincerely,

Zeljko Durdevic

***** Reviewer's comments *****

Referee #1 (Remarks for Author):

I thank the authors for their work to address the final points raised. I commend them for their efforts, particularly challenging due to the probe of human samples.

I suggest that the data on phosphorylation and ubiquitination merits publications, and should be included as supplementary material. Therefore, they may need to add the text to the manuscript.

***** Reviewer's comments *****

Referee #1 (Remarks for Author):

I thank the authors for their work to address the final points raised. I commend them for their efforts, particularly challenging due to the probe of human samples.

I suggest that the data on phosphorylation and ubiquitination merits publications, and should be included as supplementary material. Therefore, they may need to add the text to the manuscript.

We thank reviewer 1 for having evaluated our manuscript a second time. The data regarding post-transcriptional modification have been added to the manuscript as suggested. Text at line 224. Novel figures were added to the appendix

We are pleased to inform you that your manuscript is accepted for publication and is now being sent to our publisher to be included in the next available issue of EMBO Molecular Medicine.

Corresponding Author Name: Jan Rybniker

Manuscript Number: EMM-2021-14150